# Impact of Regularization on Calibration and Robustness: From the Representation Space Perspective

## Abstract

Recent studies have shown that regularization techniques using soft labels, *e.g.*, label smoothing, Mixup, and CutMix, not only enhance image classification accuracy but also mitigate miscalibration due to overconfident predictions, and improve robustness against adversarial attacks. However, the underlying mechanisms of such improvements remain underexplored. In this paper, we offer a novel explanation from the perspective of the representation space (*i.e.*, the space of the features obtained at the penultimate layer). Based on examination of decision boundaries and structure of features (or representation vectors), our study investigates confidence contours and gradient directions within the representation space. Furthermore, we analyze the adjustments in feature distributions due to regularization in relation to these contours and directions, from which we uncover central mechanisms inducing improved calibration and robustness. Our findings provide new insights into the characteristics of the high-dimensional representation space in relation to training and regularization using soft labels.

## 1 Introduction

The motivation to improve the performance of classification models has led to the development of various regularization methods that use soft labels instead of one-hot encoded hard labels for classification targets. Representative methods include label smoothing (Szegedy et al., 2016), Mixup (Zhang et al., 2018), and CutMix (Yun et al., 2019). These techniques have demonstrated significant success in improving classification accuracy across various benchmarks.

However, their impact goes beyond accuracy improvement. Studies have shown that these techniques contribute to better-calibrated models, aligning predicted probabilities more closely to actual accuracy (Guo et al., 2017; Müller et al., 2019). Furthermore, they strengthen model robustness against gradient-based adversarial attacks, where imperceptible noise is added to input data to mislead models (Goodfellow et al., 2014; Yun et al., 2019; Fu et al., 2020; Zhang et al., 2021).

While the benefits of soft labels are evident, the underlying mechanisms by which they achieve these improvements remain largely unexplained. In this paper, we offer a deeper understanding of how soft labels mitigate overconfident predictions and enhance adversarial robustness by *examining the model's representation space*. Intuitively, data points that are correctly classified with lower confidence are located near decision boundaries, making them more vulnerable to adversarial perturbations (Hein et al., 2019; Kim et al., 2024). Therefore, there exists a contradiction. If regularization alleviates overconfident predictions, features are expected to be located closer to the decision boundaries. Then, how is robustness to adversarial attacks enhanced?

To resolve this contradiction, we investigate the core mechanisms underlying model behavior in terms of calibration and robustness. For calibration, we explore the distribution of confidence contours and features, *i.e.*, the outputs of the penultimate layer within the decision boundaries. This is crucial as calibration is a measurement on the overconfidence or underconfidence of predictions. For robustness to gradient-based adversarial attacks, we focus on gradient directions and feature distributions, given that perturbations are generated based on gradients of the loss function.

We analyze the characteristics of decision boundaries, confidence contours, and gradients, in both visualizable low-dimensional representation spaces and the original high-dimensional spaces. Based on the results, we study how the feature distribution in the representation space is modified by the use of soft labels, and how such changes can improve calibration and robustness simultaneously. Our analysis spans a wide range of models in order to obtain consistent findings.

Our work can be summarized as follows:

1. Building on prior work that focused on decision boundaries, we turn our attention to confidence contours and gradient directions—the characteristics that have a direct impact on confidence calibration and robustness to gradient-based adversarial attacks. Our analysis shows that decision regions and their **confidence contours form cone-shaped structures around the origin**, while **gradients for the cross-entropy loss radiate outward from the minimal-loss point**.

2. Observing how regularization modifies the feature distribution within the representation space, we find that the magnitudes of the features are reduced, leading to tighter clustering. Using the findings above, we explain why regularization using soft labels leads to less confident predictions and improved robustness. We show that **feature vectors with smaller magnitudes improve model calibration**, as reducing the feature magnitude acts similarly to temperature scaling, a common post-hoc calibration method. Furthermore, by analyzing gradient directions in the representation space, we show that **smaller features tend to be distributed in robust regions**, which align better with the class center vector.

## 2 RELATED WORK

**Calibration and robustness.** Calibration refers to the alignment between a model's confidence and its actual accuracy. Guo et al. (2017) found that modern neural networks exhibit overconfidence, leading to miscalibrated predictions. To address this, various techniques have been proposed, including temperature scaling (Guo et al., 2017).

Simultaneously, neural networks' vulnerability to adversarial attacks—imperceptible input perturbations causing misclassification—has gained attention (Szegedy et al., 2013). An example is the Fast Gradient Sign Method (FGSM) (Goodfellow et al., 2014), which prompted extensive research into various attacks and defenses (Carlini & Wagner, 2017; Madry et al., 2018; Croce & Hein, 2020b; Deng & Mu, 2024).

**Regularization techniques.** Regularization techniques such as label smoothing, Mixup, and CutMix have been shown to enhance model generalization. Label smoothing softens targets by distributing probability mass uniformly across labels (Szegedy et al., 2016). Mixup linearly interpolates inputs and labels to create virtual training examples (Zhang et al., 2018), and CutMix replaces image regions with patches from other images, adjusting labels proportionally (Yun et al., 2019). These methods have demonstrated improvements not only in generalization but also in calibration and adversarial robustness (Yun et al., 2019; Fu et al., 2020; Zhang et al., 2021).

Several studies explored reasons behind these improvements. Thulasidasan et al. (2019) showed that data augmentation in Mixup alone using hard labels does not improve calibration, highlighting the importance of soft labels. Recent visualizations indicate that Mixup clusters data near decision boundaries, reducing model overconfidence and thus improving calibration (Fisher et al., 2024). Regarding adversarial robustness, Zhang et al. (2021) demonstrated that minimizing Mixup loss approximates minimizing adversarial loss, enhancing robustness. However, a comprehensive explanation linking soft labels to simultaneous calibration and robustness improvements from a representation perspective remains unexplored, motivating this study.

**Representation space.** Several studies explored representation spaces in deep learning using 2-dimensional visualizations, uncovering radial feature distributions (Wen et al., 2016; Wang et al., 2017; Liu et al., 2017; Chiranjeev et al., 2024). Luo et al. (2019) described cone-shaped decision regions due to radial distributions, emphasizing angularity between features. However, most studies omit the bias term in the classification layer, as the bias can be used to discriminate different classes over angularity (Wang et al., 2017). In our study, we demonstrate that these radial and cone-shaped structures persist regardless of bias terms if the number of classes is significantly smaller than the

feature dimensionality. Another notable concept is Neural Collapse, which shows that feature and weight vectors converge to an equiangular tight frame (Papyan et al., 2020). Regarding the effect of regularization on the representation space, it was observed that label smoothing brings features closer together, reducing overall prediction confidence (Müller et al., 2019). However, no prior studies have resolved the contradiction between less confident predictions and stronger robustness from such representations.

## 3 REPRESENTATION SPACE

In this section, we investigate the representation space to show how decision regions, confidence contours, and gradient directions are formed and explain the mechanisms behind the formation of these shapes and directions.

### 3.1 DECISION REGIONS

> **Key Takeaway:** Decision regions of classification models form cone-like shapes centered around the origin when the dimensionality of the representation space is sufficiently high.

A classification model typically consists of a feature extractor that maps inputs into features and a classification layer that uses those features to make decisions. The representation space of the model refers to the space where the output of the feature extractor, or more specifically, the output of the penultimate layer of the model resides (Kim et al., 2024). It usually has high dimensionality (*e.g.*, 2048 for ResNet50 (He et al., 2016) and 768 for Swin-T (Liu et al., 2021)), making it challenging to visually analyze its characteristics. To address this, we first conduct intuitive analysis by transforming the representation space to 2D, then generalize the analysis to the original space. Implementation details regarding the transformation process for visualization can be found in Appendix H.

Fig.1 visualizes ResNet50 trained on CIFAR-10 (Krizhevsky et al., 2009), comparing results with and without bias terms in the classification layer. Without biases, decision regions form perfect circular sectors (*cone-like shapes*) with radially distributed features (Fig.1a), which is consistent with prior studies (Wen et al., 2016; Luo et al., 2019; Chiranjeev et al., 2024). When biases are included, however, this pattern does not hold for certain classes (Fig. 1b), prompting some studies to exclude biases to enforce radial distributions (Wang et al., 2017; Liu et al., 2017). Nevertheless, our analysis below demonstrates that radial distributions naturally emerge when feature dimensionality is sufficiently high, regardless of bias terms.

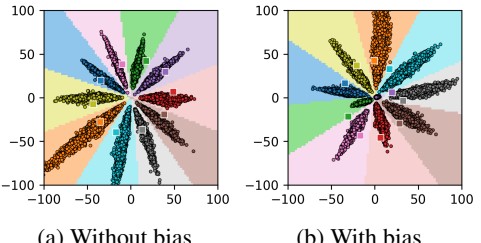

(a) Without bias  (b) With bias

Figure 1: Decision regions and feature distributions: (a) without bias in the classification layer, and (b) with bias. Without bias the regions and features are cone-shaped, radial for every class; with bias one class (purple) sits in the center with a circular region. Squares are weight vectors.

**How are decision regions shaped?** To analyze the shape of decision regions, we examine how features are processed by the classification layer, focusing on factors influencing logit calculations. This is because a feature is assigned to the class with the highest logit value. Given a feature $\mathbf{f}$, the logit $\ell_c(\mathbf{f})$ for class $c$ is expressed as follows.

$$\ell_c(\mathbf{f}) = \mathbf{w}_c^T \mathbf{f} + b_c = ||\mathbf{w}_c|| \cdot ||\mathbf{f}|| \cos\theta + b_c, \qquad (1)$$

where $\mathbf{w}_c$ and $b_c$ are the weight vector and the bias of the classification layer for class $c$, and $\theta$ is the angle between $\mathbf{f}$ and $\mathbf{w}_c$. Thus, the elements that affect prediction results are $||\mathbf{w}_c||$, $\cos\theta$, and $b_c$.

In the study of Papyan et al. (2020), it was observed that class weight norms $||\mathbf{w}_c||$ become similar, as shown in Fig. 1. Thus, without bias terms, predictions mainly depend on $\cos\theta$. In other words, decision boundaries are formed based on the degree of alignment with the weight vector, and consequently, the decision regions take the shape of cones centered at the origin. This phenomenon is observed regardless of weight initialization (see Fig. 12 in Appendix F). Further verifications of these cone-shaped regions are included in Appendix E.

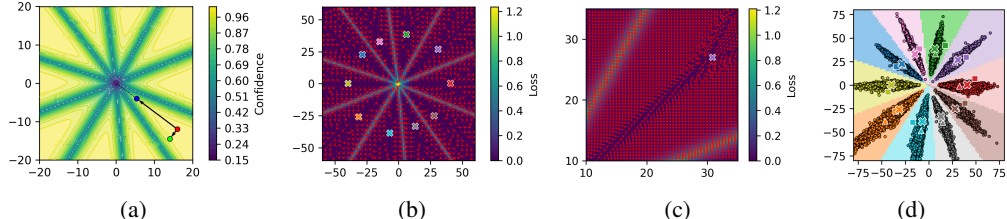

(a)              (b)              (c)              (d)

Figure 2: 2D representation space of ResNet50 on CIFAR-10. Each cross mark represents the location with the lowest loss for each class. (a) Confidence contours. (b) Loss and gradient directions. (c) Enlarged version of (b). (d) Features (circles), class means (triangles), and weight vectors (squares). A cross mark indicates the minimum loss point for each class.

For models with bias terms in the classification layer, the shape of decision regions can differ, *e.g.*, a circular decision region at the center in Fig. 1b. In this case, because bias values can be used for prediction, the purple and gray classes can be correctly classified although their weight vectors have similar directions. The appearance of a non-cone-shaped decision boundary is likely due to the difficulty that models face when trying to fit multiple classes into a cone-like structure within a constrained dimensionality, such as fitting 10 classes (or even 100 classes for CIFAR-100 (Krizhevsky et al., 2009)) into the 2D space.

Therefore, for original models, where the dimensionality of the representation space is sufficiently high, the cone shapes of the decision regions hold consistently even with the presence of bias terms. Since visual inspection is not possible in the original high-dimensional space, we take an alternative approach to verify it, whose results are presented in Tabs. 5 and 6 in Appendices C and D.

## 3.2 CONFIDENCE CONTOURS, GRADIENT DIRECTIONS

> **Key Takeaway:** Confidence contours and gradient directions radiate outward from minimal-loss points, indicating the convexity of the loss landscape.

Although research on decision boundaries has been conducted (Wen et al., 2016; Wang et al., 2017; Liu et al., 2017; Luo et al., 2019; Chiranjeev et al., 2024), the shape of confidence contours and loss gradient directions within these boundaries have been rarely examined. It is essential to study these characteristics because calibration measures how overconfident or underconfident predictions are and gradient-based adversarial attacks create perturbations based on the loss function.

**Confidence Contours**. In Fig. 2a, we present the confidence contours in each decision region in the 2D representation space. If the logit for a particular class is large in comparison to the others, the confidence in the prediction is high. From Eq. 1, the logit for a particular class is large if the feature norm is large or if the feature is well-aligned with the weight vector. This can be confirmed from the confidence contours shown in Fig. 2a. The confidence of a feature (red dot) can be lowered in two ways: 1) by moving radially toward the origin (blue dot), which reduces the feature norm, or 2) by moving toward the nearest decision boundary (green dot), which deteriorates alignment with the weight vector.

**Gradient Directions**. The cross-entropy is convex with respect to logits (Boyd & Vandenberghe, 2004). Since the logits in Eq. 1 are obtained through an affine transformation of features, the cross-entropy loss is also convex with respect to the features. To illustrate this, we examine the gradient directions in the 2D space of ResNet50 on CIFAR-10 in Figs. 2b and 2c. In areas with **high cosine similarity** to these minimal-loss points (*i.e.*, points on the line between the origin and the minimal-loss point), gradient directions point toward the origin. However, in regions with **low cosine similarity**, gradient directions appear nearly orthogonal to the direction of the origin, pointing toward the decision boundaries. Thus the gradient direction for a feature differs depending on the cosine similarity between the feature and the minimal-loss point. Overall, the gradient directions radiate outward from the minimal-loss points, highlighting the convex structure of the loss landscape. Note that, to locate these minimal-loss points, we use gradient descent.

# 4 EFFECT OF REGULARIZATION

Based on the characteristics observed in Section 3, we examine how regularization modifies feature distributions in a way to reduce overconfident predictions and enhance robustness against adversarial attacks simultaneously. Through this analysis, we aim to address the contradiction against the common expectation that if a feature with lower confidence is located closer to the decision boundary, it is likely to be more vulnerable to adversarial perturbations.

## 4.1 MEASURING FEATURE DISTRIBUTIONS

In a low-dimensional space, it is easy to identify the approximate distribution of features, as shown in Fig. 1. In a high-dimensional space, however, visualization becomes challenging, making it difficult to understand how features are distributed. Papyan et al. (2020) measured the angular distance between features and the mean of features for each class, while some others (Liu et al., 2017; Chiranjeev et al., 2024) measure angular distance of features from weight vectors. However, we argue that the angular distance should instead be measured between features and minimal-loss points.

To describe how features are distributed within the decision regions, it is natural to measure the proximity of a feature to the decision boundary based on its confidence. Following this idea, the cosine similarity of a feature with the center of the decision region of a class, noted as *class center*, should have the strongest correlation with its confidence, as the center of the decision region is expected to have the highest confidence (farthest from the decision boundary). We examine three candidates for the class center: 1) the mean of correctly classified features within a class (*class mean*), 2) the *weight vector* of the classification layer, and 3) the point where the classification loss is the lowest (*minimum loss point*).

Fig. 2d illustrates the positions of these three candidates in the 2D representation space. While the positions of class means (triangles) seem reasonable, large errors could occur if outliers exist far from the feature clusters. As seen in the pink and brown classes, the weight vectors (squares) often show a significant error, making them unsuitable as the centers of the decision regions. The minimum loss points (crosses) appear to best represent the class center. More quantitative analysis is provided in Tab. 1, where the cosine similarity between features and minimum loss points shows the highest correlation with confidence. Note that the results in Tab. 1 are from original models trained on ImageNet.

In addition, from the analysis regarding confidence in Section 3.2, features closer to the origin have lower confidence, as they are also closer to the decision boundaries. Therefore, we determine how far a specific feature is from the decision boundary using two criteria: 1) **the root mean square (RMS) of the feature** (we use RMS as the feature magnitude to compensate for different dimensionalities across models) and 2) **the cosine similarity of the feature with the *class center*** (crosses in Figs. 2b, 2c, and 2d), where the classification loss is minimal for that class.

Table 1: Pearson correlation coefficient between confidence and cosine similarity of features to class means, weight vectors, and minimum loss points. The case with the highest correlation among the three candidates is marked in bold. We omit ConvNeXt-T results trained with CutMix, as training failed to converge. See Appendix G for training details and Appendix I for experiments on hyperparameter sensitivity.

| Model | Method | Class Mean | Weight Vector | Minimum Loss Point |
|---|---|---|---|---|
| ResNet50 | Baseline | 0.35 | 0.52 | **0.56** |
| | Label smoothing | 0.36 | 0.61 | **0.70** |
| | Mixup | 0.36 | 0.60 | **0.75** |
| | CutMix | 0.39 | 0.52 | **0.65** |
| Swin-T | Baseline | 0.41 | 0.55 | **0.56** |
| | Label smoothing | 0.56 | **0.64** | **0.64** |
| | Mixup | 0.48 | 0.63 | **0.64** |
| | CutMix | 0.48 | **0.58** | **0.58** |
| MobileNetV2 | Baseline | 0.41 | 0.35 | **0.44** |
| | Label smoothing | 0.41 | 0.35 | **0.44** |
| | Mixup | 0.47 | 0.49 | **0.74** |
| | CutMix | 0.42 | 0.41 | **0.70** |
| ConvNeXt-T | Baseline | 0.26 | 0.35 | **0.37** |
| | Label smoothing | 0.73 | **0.74** | **0.74** |
| | Mixup | 0.53 | **0.57** | **0.57** |
| ViT-B-16 | Baseline | 0.24 | 0.33 | **0.34** |
| | CutMix | 0.44 | **0.49** | **0.49** |

We show the relationship of the confidence vs. the RMS of features and the cosine similarity of features with the class center in the original representation space in the top row of Fig. 3 for ResNet50. As expected, the smaller the RMS of features or the lower the cosine similarity is, the lower the

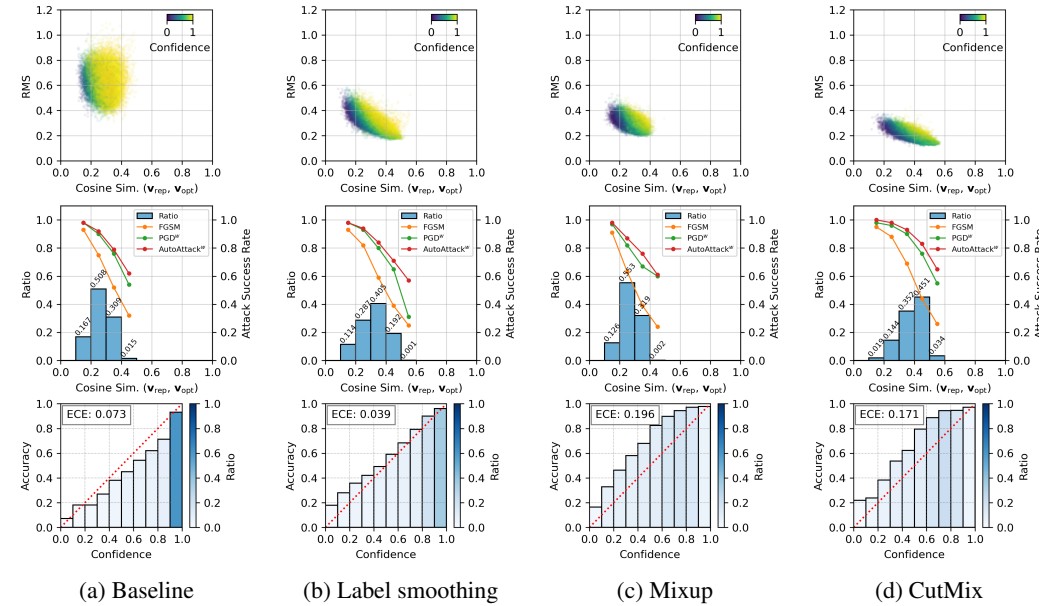

| (a) Baseline | (b) Label smoothing | (c) Mixup | (d) CutMix |

Figure 3: Evaluation results of ResNet50 on the ImageNet validation data. **Top.** Scatter plots of feature RMS and cosine similarities of features ($\mathbf{v}_{rep}$) with the class center ($\mathbf{v}_{opt}$). Colors represent confidence values. **Middle.** Histograms of cosine similarities of features to class centers, along with the attack success rates of FGSM, PGD$^w$, and AutoAttack$^w$ for each bin (hyperparameters settings for the attacks can be found in Section 4.4). For results on PGD$^s$ and AutoAttack$^s$, see Fig. 19 in Appendix K. **Bottom.** Reliability diagrams, where the transparency of bars represents the ratio of data in each confidence bin. Expected calibration error (ECE) (Guo et al., 2017) values are shown for each case.

confidence is, indicating proximity to the decision boundary. The results for Swin-T, MobileNetV2, and ConvNeXt-T can be found in Appendix K, showing similar trends.

## 4.2 IMPACT ON FEATURE DISTRIBUTIONS

> **Key Takeaway:** Regularization reduces feature norms and improves alignment with class centers, producing compact representations closer to the origin.

In Fig. 4, we present the results of training with and without regularization (label smoothing and Mixup) in the 2D representation space. See Appendix K for complete results on different regularizations and models. Training with regularization results in two notable changes. First, the RMS of features significantly decreases (Fig. 4 and the top row of Fig. 3), bringing them closer to the origin. Second, from the middle row of Fig. 3, the proportion of data with high cosine similarity between the feature and the class center increases in the regularized models. (Note that the results in Fig. 3 are from the original representation space, and these changes are consistently observed across various model architectures, as will be shown in Section 6.) Detailed analysis is as follows.

**Decrease in RMS.** The following theorem explains the phenomenon of RMS decrease due to regularization.

**Theorem 1.** *Assume* $\|\mathbf{w}_c\| \approx \|\mathbf{w}\|$ *for all classes c, a convergence behavior that prior work has demonstrated (Papyan et al., 2020; Han et al., 2021; Zhu et al., 2021). Then, the optimal solution of training* $\mathbf{f}$ *with cross-entropy using hard labels,* $\mathbf{f}^*_{hard}$, *has a larger magnitude than that using soft labels,* $\mathbf{f}^*_{soft}$, *i.e.,*

$$\|\mathbf{f}^*_{hard}\| > \|\mathbf{f}^*_{soft}\|.$$

The proof can be found in Appendix J.

To confirm this, we visualize the cross-entropy loss and gradient directions for hard and soft labels in a 2D representation space in Figs. 5a and 5b, respectively. The point with the smallest loss is marked

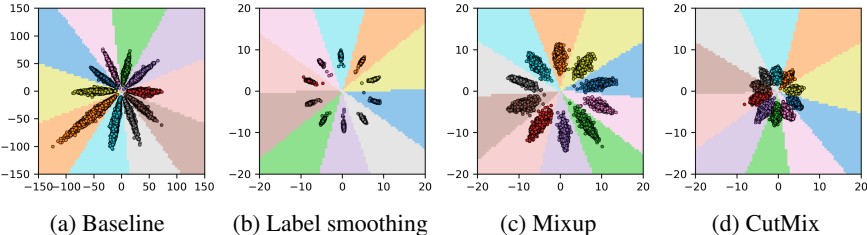

(a) Baseline      (b) Label smoothing      (c) Mixup      (d) CutMix

Figure 4: Features in the 2D representation space for ResNet50 on CIFAR-10. Note that the scales differ across figures. See Appendix E for further results.

with a white cross. We can see that for hard labels, the location of the cross mark is far from the origin, while for soft labels, it is near the origin. The same phenomenon is observed in the original representation space. In Tab. 2, the RMS values consistently decrease when soft labels are used.

**Increase in cosine similarity.** Fig. 6 shows the confidence contours in the 2D representation space of ResNet50 trained with and without regularization on CIFAR-10. For both models, we compare how well a feature needs to be aligned with the class center (crosses) to achieve a certain confidence level. Specifically, we search for features with confidence higher than 0.99 but with the worst alignment to their class center in terms of cosine similarity, in both clockwise and counterclockwise directions (red stars). By connecting these features to the origin (red lines), we can see that the angle between the lines in the regularized model is smaller (Fig. 6b). This occurs because near the origin, where the norm $||\mathbf{f}||$ is small in Eq. 1, a larger $\cos\theta$ is required to reach the same level of confidence (since confidence is positively correlated with logits). Consequently, the angular region that achieves a given confidence (*e.g.*, 0.99) becomes narrower, leading to the observed smaller angle. Therefore, features closer to the origin must be well-aligned with the class center to reach a given confidence level. Conversely, a feature located far from the origin can still achieve high confidence without being as closely aligned with the class center as a feature located near the origin (Fig. 6a). As a result, in the regularized models producing features with small RMS, the cosine similarity of features with the class center is relatively high compared to the baseline models. Tab. 2 also confirms the increase of the cosine similarity by soft labels in the original representation spaces.

### 4.3 Impact on Calibration

> **Key Takeaway:** Regularization through soft labels reduces overconfidence by scaling features closer to the origin, effectively calibrating predictions.

In the bottom row of Fig. 3, models become less confident when regularized with soft labels, reducing miscalibration due to overconfidence in the baseline training. This can be explained in relation to the RMS decrease mentioned in Section 4.2.

When the magnitude of a feature $\mathbf{f}$ is decreased (red dot becoming the blue dot in Fig. 2a) by a factor of $T$ due to training with regularization, its corresponding logit for class $c$ can be expressed as $\frac{\mathbf{w}_c^T \mathbf{f}}{T} + b_c$. Due to the cone-shaped boundaries, vectors located on the line connecting a feature and the origin are mostly classified into the same class as the feature (Tab. 5 in Appendix C). Furthermore, features with smaller RMS have lower confidence values (Section 3.2). Therefore, if the magnitude of a feature vector is scaled down and the feature moves closer to the origin, the confidence of the feature decreases, but the prediction remains unchanged. We verify that the model confidence can be reduced, or even increased by solely scaling feature vectors without compromising classification accuracy in Fig. 25 in Appendix L. This finding indicates that **decreasing magnitudes of features through soft labels enables calibration adjustment while preserving classification accuracy**.

In fact, the effect of feature scaling due to regularization is similar to the post-processing technique known as temperature scaling. Temperature scaling adjusts calibration by scaling the logit values by a factor of $T$, resulting in the logit expression $\frac{\mathbf{w}_c^T \mathbf{f} + b_c}{T}$, which is similar to the case of feature scaling. Mathematically, there is a difference of $\frac{T-1}{T} b_c$, but as discussed in Appendix D, most bias values are close to zero and do not affect the ranking of logits.

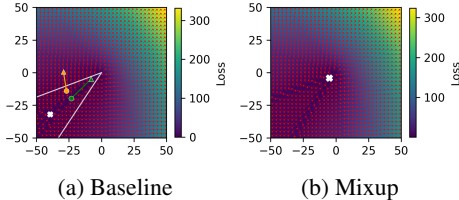 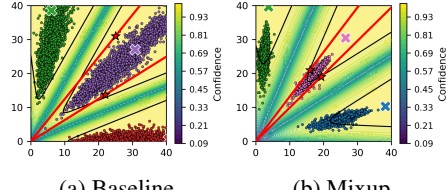

| (a) Baseline | (b) Mixup | (a) Baseline | (b) Mixup |

Figure 5: Loss and gradient directions for a certain class in the 2D representation space of ResNet50 on CIFAR-10. White crosses indicate the locations with the smallest loss. Circles and triangles represent the features of clean and perturbed data, respectively. White lines depict the decision boundary.

Figure 6: Confidence contours and features (circled dots). The 0.99 confidence contour is shown as a black line. Crosses indicate the minimum loss points. Red stars represent the features with confidence higher than 0.99 but with the poorest alignment to their class center in terms of cosine similarity; red lines connect them with the origin.

Table 2: Overall performance and feature statistics (mean and standard deviation values) across various models and training methods. Orange ECE values indicate overconfidence, while green ECE values indicate underconfidence. For PGD and AutoAttack, we present results under two hyperparameter settings, with detailed configurations provided in Section 4.4.

| Model | Method | Validation Accuracy (%) | RMS | Cosine Similarity | ECE | Attack Success Rate (%) | | | | |
|---|---|---|---|---|---|---|---|---|---|---|
| | | | | | | FGSM | PGD$^w$ | PGD$^s$ | AutoAttack$^w$ | AutoAttack$^s$ |
| ResNet50 | Baseline | 76.1 | 0.62 ± 0.09 | 0.27 ± 0.06 | 0.074 | 67.2 | 84.8 | 99.6 | 87.1 | 99.9 |
| | Label smoothing | 77.1 | 0.29 ± 0.07 | 0.32 ± 0.09 | 0.042 | 61.6 | 80.5 | 97.6 | 84.0 | 99.7 |
| | Mixup | 76.6 | 0.30 ± 0.04 | 0.27 ± 0.05 | 0.202 | 55.0 | 76.8 | 97.5 | 83.3 | 99.7 |
| | CutMix | 78.0 | 0.20 ± 0.04 | 0.38 ± 0.08 | 0.169 | 55.2 | 81.8 | 95.3 | 86.9 | 99.8 |
| Swin-T | Baseline | 75.8 | 1.38 ± 0.06 | 0.36 ± 0.09 | 0.095 | 83.9 | 85.0 | 99.7 | 85.5 | 99.8 |
| | Label smoothing | 76.3 | 0.67 ± 0.14 | 0.44 ± 0.13 | 0.033 | 79.0 | 82.4 | 99.2 | 83.6 | 99.8 |
| | Mixup | 78.2 | 0.66 ± 0.13 | 0.46 ± 0.11 | 0.013 | 71.9 | 80.9 | 98.8 | 84.9 | 98.5 |
| | CutMix | 78.7 | 0.72 ± 0.13 | 0.51 ± 0.13 | 0.050 | 77.1 | 83.2 | 99.6 | 84.8 | 99.7 |
| MobileNetV2 | Baseline | 70.8 | 0.08 ± 0.01 | 0.22 ± 0.04 | 0.081 | 83.2 | 92.7 | 99.8 | 93.4 | 99.9 |
| | Label smoothing | 71.1 | 0.08 ± 0.01 | 0.22 ± 0.04 | 0.078 | 82.3 | 91.9 | 99.8 | 92.8 | 99.9 |
| | Mixup | 70.5 | 0.04 ± 0.01 | 0.25 ± 0.05 | 0.158 | 80.3 | 89.1 | 99.6 | 92.7 | 99.8 |
| | CutMix | 70.9 | 0.03 ± 0.01 | 0.25 ± 0.06 | 0.188 | 80.3 | 91.8 | 99.7 | 93.1 | 99.9 |
| ConvNeXt-T | Baseline | 70.5 | 0.44 ± 0.08 | 0.24 ± 0.07 | 0.211 | 82.6 | 91.0 | 99.7 | 91.5 | 99.9 |
| | Label smoothing | 73.5 | 0.07 ± 0.02 | 0.49 ± 0.18 | 0.045 | 78.0 | 87.6 | 99.4 | 88.2 | 99.8 |
| | Mixup | 78.0 | 0.12 ± 0.03 | 0.28 ± 0.08 | 0.028 | 66.6 | 81.8 | 99.1 | 84.9 | 99.7 |
| ViT-B-16 | Baseline | 65.3 | 0.95 ± 0.15 | 0.23 ± 0.07 | 0.202 | 76.0 | 68.9 | 99.9 | 76.4 | 98.5 |
| | CutMix | 74.1 | 0.10 ± 0.03 | 0.40 ± 0.10 | 0.111 | 70.2 | 67.5 | 99.6 | 69.1 | 93.3 |

## 4.4 Impact on Adversarial Robustness

**Key Takeaway:** Regularization improves robustness to gradient-based adversarial attacks by enhancing feature alignment with class centers.

How does the use of regularization lead to better robustness against gradient-based adversarial attacks? To explain this, we examine the gradient directions in the 2D space of ResNet50 on CIFAR-10 in Fig. 5a. Our goal is to investigate the distance between features and the decision boundary, as well as the direction in which these features move under adversarial perturbations. To this end, we employ FGSM, which is a single-step attack that provides a clear view of how features respond to adversarial noise. Note that the gradients shown in Fig. 5a are used by FGSM to perturb the data. To visualize the gradient and perturbation directions together, we examine two sample features (green and orange) in Fig. 5a. The amount of perturbation in FGSM is set to $\epsilon = 8/255$. The feature vector well-aligned with the class center (green) remains within the decision region after perturbation, as the gradient points toward the origin. On the other hand, the feature vector poorly aligned with the class center (orange) moves along the gradient direction toward the nearby decision boundary and becomes easily misclassified. Therefore, when two features, one closely aligned with its class center and the other not, are compared, the latter is more vulnerable to attacks.

We also verify the vulnerability of features to attacks with respect to the degree of alignment in the original representation space for ResNet50 trained on ImageNet. The results are shown in the middle row of Fig. 3, where the blue bars represent the histogram of cosine similarity between features and their class centers, and the orange line shows the attack success rate of FGSM attack

for each confidence bin. It is evident that as the cosine similarity between features and class centers increases, indicating better alignment, the robustness improves.

In addition to FGSM, we evaluate model robustness under stronger adversarial attacks, such as Projected Gradient Descent (PGD) (Madry et al., 2018) and AutoAttack (Croce & Hein, 2020a). For each attack, we consider two configurations to assess the effect of perturbation strength: (i) a weaker setting (7 iterations with step size $\alpha = 0.2/255$ and perturbation limit $\epsilon = 0.4/255$ for PGD and $\epsilon = 1/255$ for AutoAttack), and (ii) a stronger setting (7 iterations with $\alpha = 2/255$, $\epsilon = 8/255$ for PGD, and $\epsilon = 8/255$ for AutoAttack, following the standard implementations). We denote these configurations as PGD$^{\text{w}}$, AutoAttack$^{\text{w}}$ and PGD$^{\text{s}}$, AutoAttack$^{\text{s}}$, respectively. The results are shown in Fig.3 and Tab.2. As shown in Fig.3 (middle row), better alignment (*i.e.*, higher cosine similarity) between features and class centers consistently enhances robustness against stronger adversarial attacks as well (see Figs. 14-24 in Appendix K for results on other models).

Thus, for a model to be robust against adversarial attacks, its features should be well-aligned with their respective class centers. **As a greater proportion of features in regularized models exhibit high cosine similarity to their class centers** (as demonstrated in Section 4.2), **these models are more robust to adversarial attacks**. For additional robustness evaluation against natural image corruptions and transfer-based black-box attacks, see Appendix B.

## 5 RESOLVING THE CONTRADICTION

Now we are ready to resolve the contradiction mentioned in the introduction. The reduction in feature RMS due to regularization shifts features closer to decision boundaries near the origin, which mitigates overconfident predictions and thereby improving calibration. Although the features become closer to the decision boundaries, the increased cosine similarity between features and class centers enhances robustness against adversarial attacks because adversarial perturbations become directed toward the origin rather than toward decision boundaries at the sides.

One may argue that because features become closer to the origin, it may be easier to perturb them so that they move across the origin into another decision region. For example, when the pink features in Figs. 4a and 4d are compared, those in Fig. 4d, being closer to the origin, could seem more susceptible to attacks, as they could be perturbed across the origin into the gray decision region. However, this is not the case because **the scale of perturbations in the representation space is not uniform but grows with the scale of the features**. Fig. 7 compares the feature RMS and perturbation RMS for various models trained on ImageNet when $\epsilon = 8/255$. The positive correlation suggests that the features close to the origin move less than the features located far from the origin under the same amount of input perturbation. Therefore, robustness is determined mostly by the direction in which a feature moves due to perturbation, rather than by the proximity of the feature to the decision boundary near the origin.

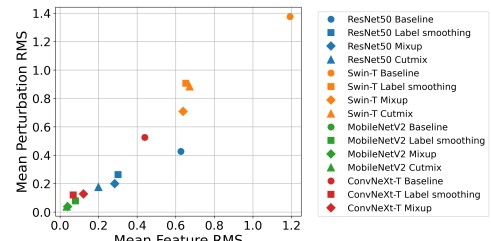

Figure 7: Feature RMS vs. perturbation RMS (the RMS of the difference between the features of clean and perturbed input images) for models trained on ImageNet.

## 6 COMPREHENSIVE EVALUATION

In Tab. 2, we present comprehensive results for various models trained with different methods on the ImageNet dataset (see Appendix G for training details). We consistently observe that, when regularization is applied, the RMS of features decreases, and the cosine similarity between features and class centers increases. These changes result in reduced overconfidence in predictions (leading even to underconfidence in some cases) and improved robustness to adversarial attacks. However, a limitation to note is that, while regularization consistently improves robustness across a variety of adversarial attacks, its effect remains limited under the strongest PGD and AutoAttack configuration. In Appendix N, we further analyze this limitation and explain why the robustness gains from soft-label regularization diminish under stronger iterative attacks. Therefore, achieving further robustness against such strong attacks likely requires complementary strategies (*e.g.*, adversarial training).

## 7 CONCLUSION

In this paper, we investigated how regularization techniques using soft labels enhance model calibration and robustness against gradient-based adversarial attacks. We analyzed decision regions, confidence contours, and gradient directions in the representation space, demonstrating that regularization reduces feature RMS and increases cosine similarity to class centers. The reduced RMS mitigates overconfident predictions, while higher cosine similarity directs perturbations toward the origin, enhancing adversarial robustness. Our findings offer new insights into the regularization dynamics in the representation space.

## 8 FUTURE WORK

An interesting direction for future work would be to examine whether other widely used regularization techniques, such as weight decay and dropout induce representation-space behaviors similar to those observed with soft-label based regularization. Understanding whether reduced feature magnitude and improved alignment arise across different regularization families would help clarify the generality of the geometric mechanisms identified in this study.

In addition, while our analysis focuses on the representation geometry induced by a standard linear classification layer trained with cross-entropy loss, it would be valuable to investigate whether the cone-shaped decision regions, feature-norm behaviors, and alignment-based robustness mechanisms persist under different learning objectives and in domains beyond vision, such as NLP, speech, or multimodal settings. Exploring how alternative losses, architectures, and pretraining schemes reshape the structure of the representation space could further broaden the applicability of our conclusions.

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

# A    DISCUSSION

Numerous studies suggest that transformer-based models outperform convolution-based models in calibration (Minderer et al., 2021) and adversarial robustness (Bai et al., 2021; Benz et al., 2021; Paul & Chen, 2022). However, these comparisons often overlook the use of soft label-based regularization during the pre-training and training phase of transformer-based models. To address this, we compare ResNet50 (25.5M parameters) and Swin-T (28.2M) using identical training conditions to achieve comparable validation accuracy. Unlike prior studies, in Tab. 2, we find that Swin-T proves to be as overconfident as ResNet50 without regularization. In addition, Swin-T is actually more susceptible to adversarial attacks, even with higher cosine similarity between features and class centers. However, because the higher cosine similarity may be due to the lower feature dimensionality of Swin-T (768) compared to ResNet50 (2048), direct comparison with respect to cosine similarity may be limited.

# B    ROBUSTNESS BEYOND GRADIENT-BASED ATTACKS

## B.1    ROBUSTNESS TO NATURAL IMAGE CORRUPTIONS

While adversarial perturbations are often constructed using gradient information, real-world inputs are more commonly degraded by natural corruption processes such as noise, blur, weather effects, or compression artifacts. Evaluating robustness under these non-adversarial perturbations is therefore essential for assessing the practical reliability of a model. To examine whether the representation-space properties induced by soft-label regularization provide benefits beyond gradient-based attacks, we evaluate all models on the ImageNet-C benchmark Hendrycks & Dietterich (2019), which consists of 15 corruption types across four categories (noise, blur, weather, and digital) with 5 severity levels for each type. The corresponding results are reported in Tab. 3.

Our goal in this experiment is twofold: (i) to measure how much the corrupted inputs degrade model accuracy relative to clean accuracy (accuracy drop), and (ii) to quantify corruption-specific degradation through corruption error (CE) and mean corruption error (mCE), following the ImageNet-C evaluation protocol.

Across a broad range of corruption categories, models trained with soft-label regularization show smaller accuracy drops and lower CE values compared to the baseline. This behavior aligns with the representation-level explanation described in Sec. 4.2. As shown in Fig. 7, inputs with smaller feature RMS experience smaller perturbations in the representation-space when corrupted. Because soft-label regularization produces features with lower RMS and stronger alignment with class centers, these models are less likely to have their representations pushed across decision boundaries under natural corruptions. This provides a coherent explanation for their improved robustness across diverse non-adversarial perturbations.

## B.2    ROBUSTNESS TO TRANSFER-BASED (BLACK-BOX) ATTACKS

Although white-box adversarial attacks provide insight into a model's local stability around inputs, real-world adversaries often lack access to model gradients or parameters. In such scenarios, attackers typically rely on *transferability*: adversarial examples crafted on a surrogate model are applied to the target model without modification. Evaluating black-box (transfer-based) robustness is therefore critical for understanding how a model behaves under more realistic adversarial conditions.

To test whether the representational properties induced by soft-label regularization also reduce adversarial transferability, we generate adversarial examples from a variety of surrogate models and apply them to each target model. We measure the attack success rate under FGSM, PGD, and AutoAttack, following standard transfer-based attack protocols, and report the results in Tab 4.

Consistent with white-box attack results (Tab. 2), models trained with soft-label based regularization exhibit significantly reduced transfer attack success rates. This observation aligns with prior work showing that transferability arises from shared adversarially vulnerable directions or overlapping gradient subspaces across models (Tramèr et al., 2017; Adam et al., 2019; Yang et al., 2021). In our case, lower RMS and stronger class-center alignment, which are key properties of soft-label regularization, cause perturbed features to move toward the origin rather than toward decision boundaries.

Table 3: ImageNet-C robustness evaluation. 'Acc. Drop' denotes the decrease in accuracy on corrupted inputs relative to clean accuracy. 'Error' refers to the error rate on clean data. The values for each corruption type on the right represent corruption errors (CE) normalized relative to the AlexNet baseline, following the original implementation (Hendrycks & Dietterich, 2019). 'mCE' represents the mean CE across all corruption categories. 'LS.' denotes label smoothing. Overall, models trained with soft-label based regularization exhibit reduced accuracy drop under corruption.

| Model | Method | Acc. Drop | Error | mCE | Noise | | | Blur | | | | Weather | | | | Digital | | | |
|---|---|---|---|---|---|---|---|---|---|---|---|---|---|---|---|---|---|---|---|
| | | | | | Gauss. | Shot | Impulse | Defocus | Glass | Motion | Zoom | Snow | Frost | Fog | Bright | Contrast | Elastic | Pixel | JPEG |
| AlexNet | - | 62.7 | 43.5 | 100 | 100 | 100 | 100 | 100 | 100 | 100 | 100 | 100 | 100 | 100 | 100 | 100 | 100 | 100 | 100 |
| ResNet50 | Baseline | 50.1 | 23.9 | 78.4 | 77.1 | 77.9 | 79.7 | 77.3 | 89.0 | 82.3 | 80.3 | 83.3 | 78.1 | 74.0 | 63.0 | 75.7 | 87.8 | 72.8 | 77.3 |
| | LS. | 49.2 | 22.9 | 76.7 | 77.1 | 79.3 | 80.4 | 74.9 | 88.4 | 80.5 | 78.2 | 79.5 | 76.8 | 71.4 | 60.5 | 74.1 | 84.7 | 70.5 | 74.6 |
| | Mixup | 41.7 | 23.4 | 70.5 | 63.2 | 66.1 | 65.0 | 79.1 | 90.7 | 81.7 | 77.7 | 70.7 | 57.3 | 53.1 | 59.6 | 55.9 | 89.0 | 70.3 | 77.4 |
| | CutMix | 53.3 | 22.0 | 80.1 | 77.4 | 78.9 | 81.1 | 88.9 | 97.7 | 91.0 | 92.0 | 79.4 | 78.2 | 68.9 | 60.9 | 73.5 | 86.7 | 72.1 | 75.5 |
| MobilenetV2 | Baseline | 58.9 | 29.2 | 89.7 | 89.7 | 90.6 | 88.9 | 88.5 | 97.5 | 90.1 | 91.1 | 91.8 | 92.5 | 85.3 | 77.9 | 86.6 | 96.1 | 89.7 | 89.3 |
| | LS. | 58.9 | 28.9 | 89.6 | 90.5 | 91.6 | 89.1 | 88.5 | 97.5 | 87.8 | 87.5 | 91.0 | 92.1 | 85.0 | 77.9 | 86.2 | 96.6 | 91.1 | 91.4 |
| | Mixup | 53.0 | 29.5 | 85.0 | 80.9 | 84.1 | 83.3 | 90.6 | 100.4 | 93.3 | 90.7 | 81.3 | 73.6 | 70.8 | 73.1 | 72.7 | 96.1 | 92.2 | 91.6 |
| | CutMix | 57.0 | 29.1 | 87.9 | 92.0 | 93.2 | 92.4 | 78.5 | 86.4 | 76.8 | 82.2 | 91.7 | 93.8 | 86.2 | 77.9 | 88.0 | 97.8 | 90.1 | 91.8 |
| Swin-T | Baseline | 51.2 | 24.2 | 79.4 | 81.0 | 82.6 | 81.4 | 78.0 | 84.9 | 76.1 | 81.6 | 84.5 | 83.7 | 78.0 | 66.0 | 77.7 | 82.2 | 76.0 | 77.9 |
| | LS. | 49.7 | 23.7 | 77.6 | 80.8 | 82.3 | 81.1 | 74.3 | 86.0 | 72.3 | 76.7 | 81.5 | 81.9 | 75.7 | 64.6 | 79.2 | 78.6 | 74.1 | 75.1 |
| | Mixup | 42.4 | 21.8 | 69.7 | 68.7 | 71.7 | 67.7 | 75.7 | 85.6 | 73.3 | 78.7 | 67.1 | 58.0 | 53.4 | 58.8 | 58.1 | 78.2 | 77.9 | 72.5 |
| | CutMix | 53.1 | 21.3 | 79.4 | 82.1 | 83.7 | 81.5 | 85.0 | 93.0 | 85.6 | 91.2 | 78.9 | 78.8 | 71.1 | 60.2 | 74.1 | 78.0 | 74.5 | 73.3 |
| ConvNeXt-T | Baseline | 58.2 | 29.5 | 89.2 | 91.2 | 91.7 | 90.2 | 85.0 | 93.0 | 85.6 | 91.2 | 91.1 | 92.3 | 87.7 | 80.9 | 87.3 | 92.3 | 86.9 | 91.8 |
| | LS. | 53.9 | 26.5 | 83.5 | 86.7 | 87.2 | 86.1 | 80.4 | 90.3 | 80.8 | 85.3 | 85.6 | 85.9 | 80.5 | 71.7 | 81.7 | 86.7 | 78.8 | 84.5 |
| | Mixup | 40.7 | 22.0 | 68.4 | 63.5 | 67.9 | 64.2 | 74.3 | 87.0 | 70.1 | 71.7 | 67.0 | 56.5 | 55.1 | 59.1 | 56.5 | 83.9 | 76.0 | 73.1 |

Table 4: Transfer-based (black-box) attack success rates (ASR) generated from surrogate models. 'LS.' denotes label smoothing. Consistent with the white-box ASR, soft-label based regularizations significantly reduce the transferability of adversarial perturbations compared to baseline models.

| Surrogate Model | | | ResNet50 | | | | Swin-T | | | |
|---|---|---|---|---|---|---|---|---|---|---|
| Model | Attack Type | Method | Baseline | LS. | Mixup | CutMix | Baseline | LS. | Mixup | CutMix |
| MobileNetV2 | FGSM | Baseline | 25.8 | 24.3 | 18.0 | 23.2 | 28.4 | 26.3 | 18.3 | 25.8 |
| | | Mixup | 21.8 | 20.8 | 18.3 | 20.9 | 26.8 | 24.1 | 18.5 | 24.4 |
| | PGD$^S$ | Baseline | 20.5 | 19.5 | 13.5 | 18.7 | 21.7 | 19.5 | 13.9 | 19.9 |
| | | Mixup | 13.3 | 12.7 | 11.8 | 12.9 | 16.1 | 14.0 | 11.8 | 14.5 |
| | AutoAttack$^S$ | Baseline | 18.0 | 16.9 | 11.4 | 16.8 | 19.8 | 17.4 | 11.3 | 18.7 |
| | | Mixup | 8.6 | 8.1 | 7.3 | 8.3 | 11.4 | 9.9 | 7.7 | 10.3 |
| ConvNeXt-T | FGSM | Baseline | 28.8 | 27.1 | 21.4 | 26.7 | 40.8 | 37.6 | 29.2 | 37.8 |
| | | Mixup | 23.7 | 23.1 | 21.5 | 23.1 | 28.0 | 25.4 | 23.1 | 27.1 |
| | PGD$^S$ | Baseline | 21.1 | 19.9 | 15.1 | 19.8 | 36.6 | 31.5 | 24.2 | 33.7 |
| | | Mixup | 15.4 | 14.8 | 15.1 | 15.1 | 18.3 | 15.5 | 15.6 | 17.6 |
| | AutoAttack$^S$ | Baseline | 19.2 | 18.1 | 13.1 | 17.8 | 37.6 | 32.3 | 24.1 | 35.4 |
| | | Mixup | 11.1 | 10.7 | 10.0 | 10.5 | 14.1 | 11.6 | 10.4 | 12.8 |

As a result, transferred adversarial examples generated on a surrogate model are less effective at crossing the target model's decision boundaries. This explains the consistently lower attack transferability and stronger black-box robustness observed in regularized models.

## C    DECISION REGIONS OF ORIGINAL MODELS

In this section, we investigate whether, when the dimensionality of the representation space is sufficiently high (*e.g.*, 2048 for ResNet50), the decision regions have cone shapes regardless of the presence of bias terms. One simple way to verify this is to gradually move a correctly classified feature linearly toward the origin and observe when it becomes misclassified for the first time. This process is illustrated in Fig. 8. If the decision regions are cone-shaped, the classification result will remain consistent until the feature arrives at the origin. Actually, the intersection point of the cone-shaped decision regions does not precisely coincide with the origin, but is close to the origin. Thus, the moment that the misclassification occurs will be only at the final stage of the linear movement. On the other hand, if the regions are not cone-shaped, meaning another class region lies between the feature and the origin, the feature will become misclassified early during the movement.

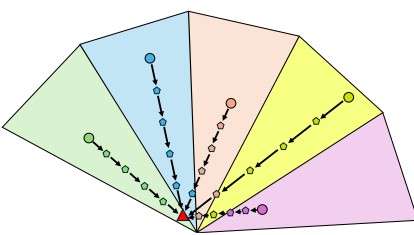

Figure 8: Illustration of linear movements of features (circled dots) toward the origin (red triangle). Each large triangular region represents the decision region of a specific class. Pentagons represent the intermediate positions of features as they move toward the origin. The colors within each dot indicate the class to which they are classified.

Table 5: Accuracy and the mean ($\pm$ standard deviation) of the first movement index of misclassification for various models on different datasets.

| Model | Dataset | Accuracy (%) | Index |
|---|---|---|---|
| ResNet50 | CIFAR-10 | 92.5 | 99.9 ±1.6 |
| | CIFAR-100 | 71.4 | 99.4 ±3.9 |
| | ImageNet | 76.1 | 99.8 ±1.9 |
| Swin-T | CIFAR-10 | 89.3 | 99.9 ±1.3 |
| | CIFAR-100 | 66.7 | 99.6 ±2.7 |
| | ImageNet | 75.8 | 99.3 ±3.2 |
| MobileNetV2 | CIFAR-10 | 92.6 | 99.8 ±1.7 |
| | CIFAR-100 | 71.7 | 98.9 ±4.2 |
| | ImageNet | 70.8 | 91.3 ±10.7 |
| ConvNeXt-T | CIFAR-10 | 94.5 | 99.7 ±2.5 |
| | CIFAR-100 | 70.6 | 98.1 ±5.2 |
| | ImageNet | 70.5 | 93.9 ±8.1 |

We verify this for ResNet50, Swin-T, MobileNetV2 (Howard, 2017), and ConvNeXt-T (Liu et al., 2022) on the test sets of CIFAR-10 and CIFAR-100, and the validation set of ImageNet (Russakovsky et al., 2015). For each feature in the representation space, we linearly move it toward the origin over 100 uniform steps. If the index of the first misclassified step is close to 100, it suggests that the decision region is likely cone-shaped. The results are shown in Tab. 5. Since all indices are over 91 on average, we can confirm that decision regions are divided into cone shapes even with the presence of bias terms, if the dimensionality of the representation space is high enough. Further verification can be found in Appendix D.

## D  BIAS TERM IN THE CLASSIFICATION LAYER

In this section, we provide a more detailed discussion on the effects of the bias term on decision regions, as introduced in Section 3.1, both in the 2D and original representation spaces.

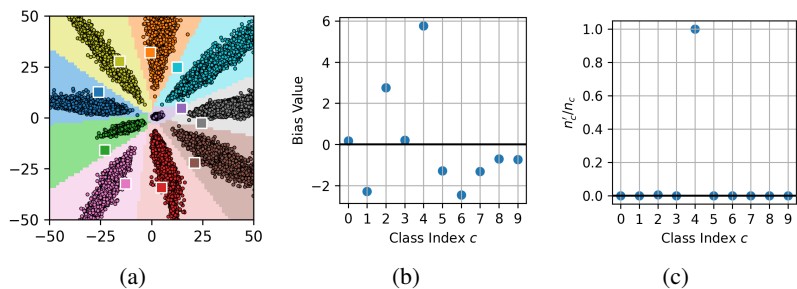

(a)  (b)  (c)

Figure 9: Results for ResNet50 with a 2D representation space trained on CIFAR-10. (a) 2D representation space. Circled and squared dots represent the features and weight vectors, respectively. Different colors indicate different class regions and classification results. (b) Bias values for each class. (c) $n'_c/n_c$ values for each class.

### D.1  2D REPRESENTATION SPACES

Fig. 9a shows the 2D representation space of ResNet50 with the classification layer bias (identical to Fig. 1b), trained on CIFAR-10. While most classes form cone-shaped decision regions, a purple class with a circular decision region appears near the origin. As discussed in Section 3.1, cone-shaped decision regions arise because the classification result is determined by the weight vector that is aligned most closely with the feature vector. However, when two weight vectors have similar directions, as the purple and gray classes in Fig. 9a, the final classification is determined by the biases. To validate this, we examine the bias values for all 10 classes in Fig. 9b. It is clear that

the purple class (class 4) has a significantly higher bias value compared to the other classes. This suggests that without the bias, features from class 4 would not be correctly classified.

We calculate the ratio of instances where prediction results depend on the bias values when determining logits. To elaborate, for an arbitrary class $c$, we count the number of correctly classified samples $n_c = \sum_{i=1}^{N} \mathbf{1}\left(\hat{y}_i = y_i = c\right)$, where $\hat{y}_i$ is the predicted class, $y_i$ is the true class for sample $i$, and $N$ is the total number of samples. Then, let $\mathbb{A}_c$ be the set of indices of samples that are correctly predicted into class $c$. Among such samples, we count the number of samples $n'_c$ where the prediction results would change if the logits are calculated without biases. This can be expressed as $n'_c = \sum_{j \in \mathbb{A}_c} \mathbf{1}\left(\hat{y}_j^{\text{no bias}} \neq \hat{y}_j\right)$, where $\hat{y}_j^{\text{no bias}}$ is the predicted class for sample $j$ when logits are calculated without biases. Therefore, if the ratio $n'_c/n_c$ is large, the presence of bias values are crucial for the data correctly classified as class $c$.

Fig. 9c shows the value of $n'_c/n_c$ for each class. The classes with cone-shaped decision regions have low $n'_c/n_c$ values, indicating that the bias terms are not necessary for correctly classifying these classes. However, for class 4 (the purple class in Fig. 9a), the value of $n'_c/n_c$ is 1, suggesting that non-cone-shaped decision regions rely on the bias for accurate classification. Therefore, by examining $n'_c/n_c$, we can determine whether the bias is needed for correct classification of a particular class and infer the shape of its decision region in the representation space.

More examples on the effect of the bias in the 2D representation space are provided in Appendix E.

## D.2 ORIGINAL REPRESENTATION SPACES

Now we verify that the bias-dependence phenomenon rarely occurs in the original high-dimensional representation spaces. Tab. 6 shows the mean and standard deviation of $|b_c|$ (the absolute bias values for each class $c$) and the ratios $n'_c/n_c$ for the models trained on ImageNet. The small $|b_c|$ values indicate minimal dependency of the classification results on the bias, leading to the small $n'_c/n_c$ values. Consequently, cone-shaped decision regions are formed for all classes.

Table 6: Mean, standard deviation, and maximum values of $|b_c|$ and $n'_c/n_c$ for models trained on ImageNet.

| Model | Method | $\|b_c\|$ **Mean (±std)** | $n'_c/n_c$ **Mean (±std)** |
|---|---|---|---|
| ResNet50 | Baseline | 0.009 (±0.007) | 0.0005 (±0.005) |
| | Label smoothing | 0.011 (±0.008) | 0.0007 (±0.005) |
| | Mixup | 0.008 (±0.006) | 0.0005 (±0.005) |
| | CutMix | 0.009 (±0.007) | 0.0003 (±0.003) |
| Swin-T | Baseline | 0.029 (±0.022) | 0.0010 (±0.006) |
| | Label smoothing | 0.015 (±0.010) | 0.0009 (±0.006) |
| | Mixup | 0.026 (±0.020) | 0.0015 (±0.008) |
| | CutMix | 0.026 (±0.021) | 0.0013 (±0.007) |
| MobileNetV2 | Baseline | 0.339 (±0.254) | 0.0208 (±0.036) |
| | Label smoothing | 0.331 (±0.251) | 0.0219 (±0.041) |
| | Mixup | 0.264 (±0.184) | 0.0246 (±0.047) |
| | CutMix | 0.262 (±0.190) | 0.0245 (±0.045) |
| ConvNeXt-T | Baseline | 0.463 (±0.349) | 0.0097 (±0.026) |
| | Label smoothing | 0.212 (±0.084) | 0.0022 (±0.009) |
| | Mixup | 0.362 (±0.279) | 0.0165 (±0.036) |
| MobileNetV2 | PyTorch V1 | 0.028 (±0.022) | 0.0037 (±0.014) |
| | PyTorch V2 | 0.053 (±0.042) | 0.0084 (±0.023) |
| EfficientNet-B1 | PyTorch V1 | 0.054 (±0.041) | 0.0037 (±0.015) |
| | PyTorch V2 | 0.116 (±0.084) | 0.0065 (±0.021) |
| ViT-B/16 | PyTorch Swag Linear V1 | 0.030 (±0.026) | 0.0022 (±0.011) |
| | PyTorch V1 | 0.016 (±0.013) | 0.0007 (±0.005) |

## E MORE VISUALIZATION OF 2D AND 3D REPRESENTATION SPACES

Here, we present additional examples of decision regions divided around the origin, extending the discussion in Section 3.1. Fig. 10 shows the visualization results for ResNet50 and Swin-T on the CIFAR-10 dataset, when the models are trained without the bias in the classification layer. Regardless of the differences in model structures, it can be observed that the decision regions are divided into circular sectors, *i.e.*, *cone-like shapes*, centered around the origin, with features radially dis-

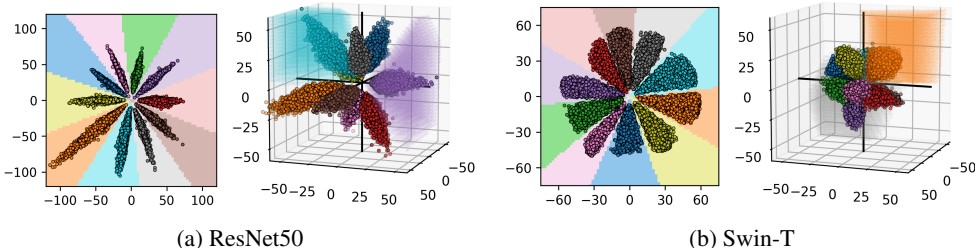

(a) ResNet50               (b) Swin-T

Figure 10: Visualization of the 2D and 3D representation spaces of ResNet50 and Swin-T on CIFAR-10. Circled dots represent output features, with different colors indicating different classes. The whole 2D planes are also colored according to the classification result of each point in the plane. In the case of 3D, regions corresponding only two classes are colored as examples for the sake of visualization. The values of the feature vectors are used as coordinates for the x, y, and z axes. Note that the scales differ across figures to best visualize the representation spaces.

tributed within these regions. Note that for the results shown in Fig. 10, regularization using soft labels is not applied.

Fig. 11 provides further examples on how regularization and classification layer bias influence the decision regions and feature distributions in the 2D representation space. As discussed in Section 3.1 and Appendix D, when the bias term is present in the visualizable low-dimensional representation space, some decision regions deviate from cone shapes, as the bias value may have a greater effect than the angular differences between features. Regarding the effect of regularization, as explained in Section 4.2, the use of soft labels leads to a significant reduction in the magnitude of the features.

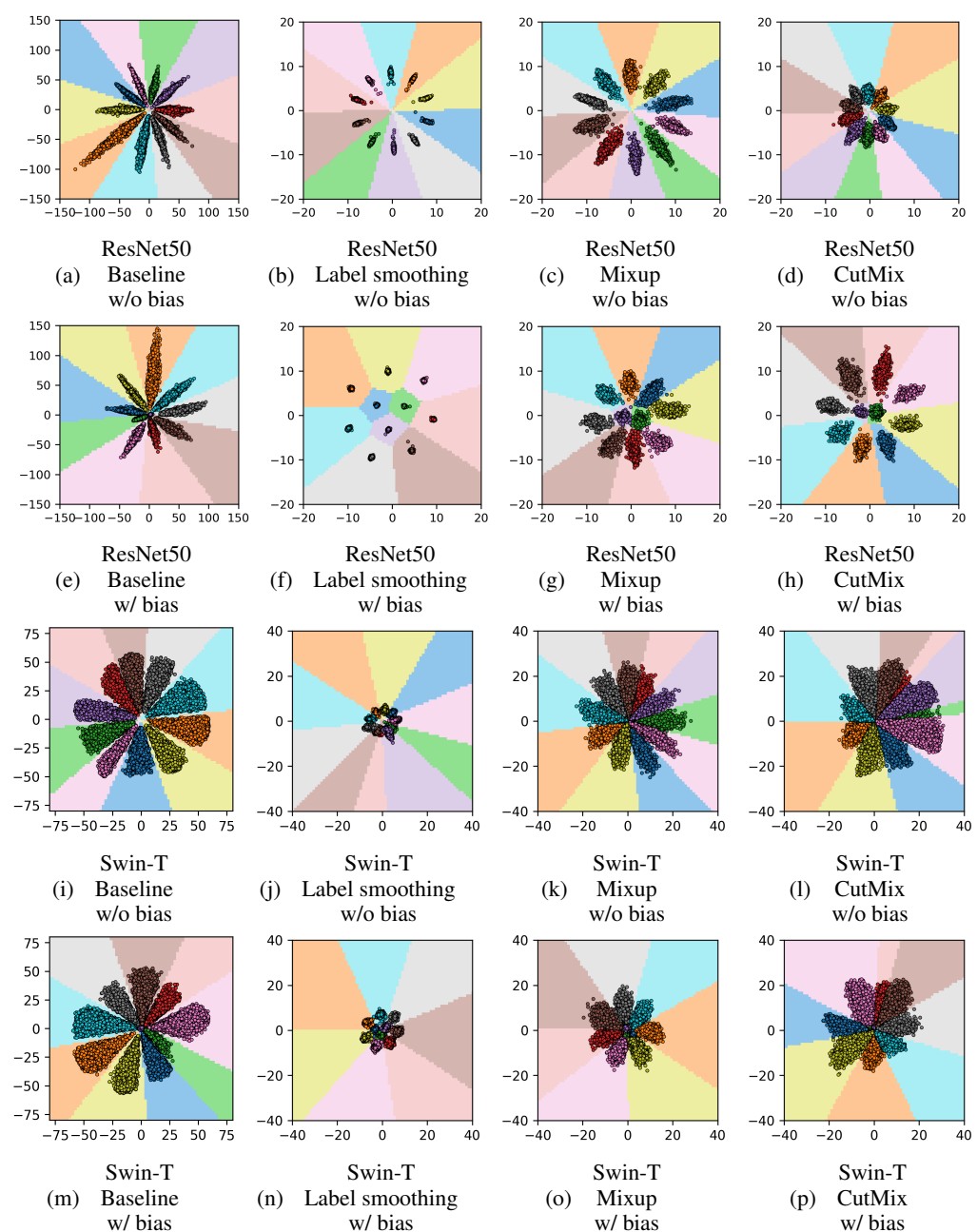

Figure 11: Decision regions and feature distribution in the 2D representation space for ResNet50 and Swin-T on CIFAR-10. Note that the scales differ across figures.

## F    EFFECT OF WEIGHT INITIALIZATION

In this section, we verify whether the formation of cone-shaped decision regions around the origin depends on weight initialization. Fig. 12 shows the process of how the decision regions change during training of ResNet50 (with the classification layer bias) on CIFAR-10 with four different weight initializations (implementation details can be found in Appendix G). In Fig. 12a, we use the default Kaiming uniform weight initialization (He et al., 2015). Before training, the features are distributed close to the origin, as the model's weights are initialized as small values; some decision regions are divided around it, while for the other decision regions, due to random weight initialization, they have not yet been assigned to regions in the representation space. In Fig. 12b, we set the initial weights

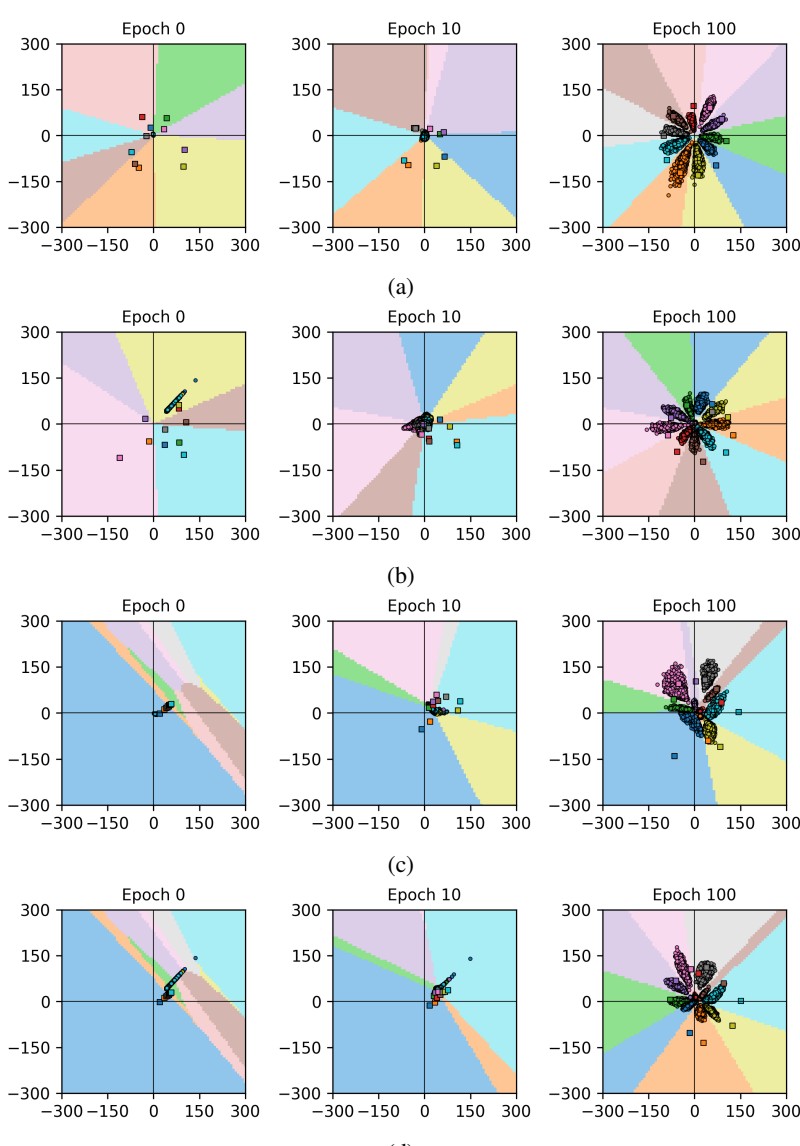

Figure 12: Changes in the 2D representation space as training progresses for ResNet50 on CIFAR-10 with different weight initializations. Features and weight vectors are represented as circles and squares, respectively (weight vectors are scaled to indicate directions).

of the feature extractor in such a way that the features are positioned far from the origin while the weights of the classification layer are initialized as in Fig. 12a so that the decision regions remain divided around the origin. In Fig. 12c, we set the initial weights of the classification layer so that the decision regions are divided far from the origin (but the initial features are located around the origin due to the default initialization of the feature extractor). Finally, in Fig. 12d, we adjust the initial weights of the entire model so that the center of the decision regions is shifted and the features are distributed far from the origin. It can be observed from the figure that in all four cases, the decision regions eventually evolve into cone-like shapes around the origin as training progresses.

## G IMPLEMENTATION DETAILS

Here, we provide the implementation and training details for the models used in the main paper and Appendix.

### G.1 2D AND 3D REPRESENTATION SPACES

We train models on the CIFAR-10 dataset for 100 epochs using the SGD optimizer with a momentum of 0.9 and a weight decay of 0.0001. For learning rate scheduling, we apply a linear warmup from 0 to 0.01 over the first 10 epochs, followed by a cosine annealing scheduler for the remaining 90 epochs. Regarding regularization hyperparameters, we set the label smoothing value to 0.1 and use an alpha value of 0.2 for both Mixup and CutMix. However, for Mixup in Figs. 4d, 5b, and 6b, the alpha value is set to 0.05 for better visualization.

Regarding weight initialization in Fig. 12, to initialize the model to distribute features far from the origin, we first train the feature extractor alone, using the L2 norm between the extracted features and $\mathbf{v} = [80, 80]$ as the loss function for 1 epoch using the SGD optimizer with a learning rate of 0.0005. After this, we attach a linear classifier and train the full model using the training settings listed above. To initialize the classifier so that the decision regions are divided far from the origin, we use hand-crafted weight matrix $\mathbf{W}$ and bias vector $\mathbf{b}$ in the classification layer, which are given as follows:

$$
\mathbf{W} = \begin{bmatrix} 0.1318 & -0.0165 \\ 0.2245 & 0.0709 \\ 0.2630 & 0.1030 \\ 0.2881 & 0.1138 \\ 0.2947 & 0.1381 \\ 0.3189 & 0.1362 \\ 0.3195 & 0.1567 \\ 0.3323 & 0.1654 \\ 0.3482 & 0.1684 \\ 0.3619 & 0.1806 \end{bmatrix}, \quad \mathbf{b} = \begin{bmatrix} 15.705 \\ 9.045 \\ 5.644 \\ 3.144 \\ 0.834 \\ -1.286 \\ -3.442 \\ -5.799 \\ -8.489 \\ -11.972 \end{bmatrix}.
$$

### G.2 ORIGINAL REPRESENTATION SPACES

We train models on the CIFAR-10, CIFAR-100, and ImageNet datasets for 300 epochs using the AdamW optimizer (Loshchilov, 2017) with a weight decay of 0.0005. For learning rate scheduling, we apply a linear warmup from 0 to 0.001 over the first 20 epochs, followed by a cosine annealing scheduler for the remaining 280 epochs. When training Swin-T on CIFAR-10 and CIFAR-100, we increase the learning rate from 0 to 0.01 during the warmup phase to improve test accuracy. Regarding the regularization hyperparameters, we set the label smoothing value to 0.1 and use an alpha value of 1.0 for both Mixup and CutMix.

We trained models using a 12th Gen Intel® Core™ i7-12700K CPU and a NVIDIA RTX A6000 GPU. Specifically, training MobileNetV2 on the ImageNet dataset for 300 epochs required approximately 7 days.

The weights used for MobileNetV2, EfficientNet-B1 (Tan, 2019), and ViT-B/16 (Dosovitskiy et al., 2021) in Tabs. 6, 7, and 8 are pretrained weights from PyTorch. In the PyTorch framework, these pretrained weights can be accessed using the names `IMAGENET1K_V1`, `IMAGENET1K_V2`, and `IMAGENET1K_SWAG_LINEAR_V1`. Details on the regularization hyperparameters (using soft labels) used to train these weights can be found in Tab. 7.

## H REPRESENTATION SPACE VISUALIZATION

To visualize the representation space in 2D (or 3D), we insert a linear layer between the feature extractor and the classification layer, mapping the output features into 2D (or 3D) vectors, and adjust the input dimension of the classification layer accordingly. As a result, the new representation space is 2D (or 3D), which facilitates visual examination, as in Figs. 1, 2d, 4, 9a, 10, 11, and 12. To

Table 7: Regularization hyperparameters to train models on ImageNet.

| Model | Method | Label smoothing | Mixup | CutMix |
|---|---|---|---|---|
| MobileNetV2 | PyTorch V1 | - | - | - |
| | PyTorch V2 | 0.10 | 0.2 | 1.0 |
| EfficientNet-B1 | PyTorch V1 | - | - | - |
| | PyTorch V2 | 0.10 | 0.2 | 1.0 |
| ViT-B/16 | PyTorch Swag Linear V1 | - | 0.1 | - |
| | PyTorch V1 | 0.11 | 0.2 | 1.0 |

Table 8: Overall performance and feature statistics (mean and standard deviation values) across various models and training methods. Orange ECE values indicate overconfidence, while green ECE values indicate underconfidence. For PGD and AutoAttack, we present results under two hyperparameter settings, with detailed configurations provided in Section 4.4.

| Model | Method | Validation Accuracy (%) | RMS | Cosine Similarity | ECE | Attack Success Rate (%) | | | | |
|---|---|---|---|---|---|---|---|---|---|---|
| | | | | | | FGSM | PGD$^w$ | PGD$^s$ | AutoAttack$^w$ | AutoAttack$^s$ |
| MobileNetV2 | PyTorch V1 | 71.9 | 0.78 ±0.09 | 0.31 ±0.07 | 0.028 | 85.7 | 77.6 | 94.3 | 90.7 | 99.8 |
| | PyTorch V2 | 72.0 | 0.28 ±0.05 | 0.36 ±0.09 | 0.367 | 73.2 | 64.6 | 88.2 | 90.2 | 99.8 |
| EfficientNet-B1 | PyTorch V1 | 77.6 | 0.34 ±0.08 | 0.32 ±0.09 | 0.091 | 65.1 | 48.8 | 81.4 | 69.3 | 99.6 |
| | PyTorch V2 | 78.9 | 0.15 ±0.02 | 0.34 ±0.07 | 0.271 | 62.1 | 46.2 | 78.4 | 65.0 | 99.1 |
| ViT-B/16 | PyTorch Swag Linear V1 | 81.8 | 1.28 ±0.08 | 0.33 ±0.08 | 0.018 | 58.8 | 48.5 | 79.5 | 71.3 | 99.7 |
| | PyTorch V1 | 81.1 | 0.56 ±0.09 | 0.57 ±0.11 | 0.055 | 54.9 | 37.6 | 65.2 | 51.3 | 98.5 |

distinguish between different decision regions in the 2D (or 3D) representation space, we input a 2D grid (or 3D cube) with a fixed range into the classification layer. Each point in the grid or cube is then classified into a specific class. We visualize the decision regions by coloring each point in the grid or cube according to its predicted class. Next, using the values of the 2D (or 3D) feature vectors as coordinates, we plot their locations in the representation space, marking them with black-bordered circles colored by their predicted class.

# I  HYPERPARAMETER SENSITIVITY ANALYSIS OF CLASS CENTERS

In this section, we examine the hyperparameter sensitivity of the class centers defined in Section 4.2, where the class center is given by the minimum-loss point in the representation space. Since these centers are obtained through gradient-based optimization, it is important to verify that their locations are stable under different optimization settings and are not artifacts of a particular choice of hyperparameters.

To assess this, we conduct additional experiments using the ResNet-50 model trained under the baseline setting on ImageNet, and vary the optimization configuration used to locate the minimum-loss centers. Specifically, we use SGD with and without a cosine-annealing learning-rate scheduler, test learning rates of 0.001, 0.01, and 0.1, and vary the number of gradient-descent iterations across 50, 100, and 200 steps. This results in 18 distinct optimization configurations.

Across all settings, the resulting Pearson correlations between confidence and cosine similarity to the estimated centers remain highly consistent, ranging only from 0.5353 to 0.5452. This extremely small variation demonstrates that the estimated class centers are stable across a wide range of hyperparameters and that center-estimation noise does not materially influence the quantitative findings reported in the main paper. The full results are presented in Table 9.

Table 9: Stability of the minimum-loss center across optimization hyperparameters. 'Corr. Coeff.' denotes the Pearson correlation coefficient between model confidence and the cosine similarity to the estimated minimum-loss class center. Across all hyperparameter combinations, the coefficients remain highly stable, indicating robust center estimation.

| Optimizer | Scheduler | Learning Rate | Iterations | Corr. Coeff. |
|---|---|---|---|---|
| SGD | None | 0.001 | 50 | 0.5353 |
| | | | 100 | 0.5368 |
| | | | 200 | 0.5410 |
| | | 0.01 | 50 | 0.5409 |
| | | | 100 | 0.5428 |
| | | | 200 | 0.5452 |
| | | 0.1 | 50 | 0.5396 |
| | | | 100 | 0.5396 |
| | | | 200 | 0.5396 |
| | Cosine Ann. | 0.001 | 50 | 0.5349 |
| | | | 100 | 0.5354 |
| | | | 200 | 0.5370 |
| | | 0.01 | 50 | 0.5386 |
| | | | 100 | 0.5412 |
| | | | 200 | 0.5430 |
| | | 0.1 | 50 | 0.5394 |
| | | | 100 | 0.5395 |
| | | | 200 | 0.5396 |

## J PROOF OF THEOREM 1

Let $y$ be the ground-truth class from the set of classes $c \in \{1, \ldots, K\}$. We define the logit for class $c$ as follows,

$$z_c = \mathbf{w}_c^\top \mathbf{f} + b_c = \|\mathbf{w}\| \cdot \|\mathbf{f}\| \cos \theta_c + b_c, \tag{2}$$

where $\theta_c = \angle(\mathbf{w}_c, \mathbf{f})$. Let the target vector be $\mathbf{t} = (t_1, \ldots, t_K)$, where $t_y > \max_{c \neq y} t_c$ and $\sum_{c=1}^{K} t_c = 1$. The training objective to minimize cross-entropy can be written as

$$\min \mathrm{CE} = \min -\sum_{c=1}^{K} t_c \log \frac{\exp(z_c)}{\sum_{j=1}^{K} \exp(z_j)} \tag{3}$$

$$= \min -\sum_{c=1}^{K} \log \left( \frac{\exp(z_c)}{\sum_{j=1}^{K} \exp(z_j)} \right)^{t_c} \tag{4}$$

$$= \min -\prod_{c=1}^{K} \left( \frac{\exp(z_c)}{\sum_{j=1}^{K} \exp(z_j)} \right)^{t_c} \tag{5}$$

$$= \min -\prod_{c=1}^{K} \frac{\exp(t_c z_c)}{\left( \sum_{j=1}^{K} \exp(z_j) \right)^{t_c}} \tag{6}$$

$$= \min -\frac{\exp(\sum_{c=1}^{K} t_c z_c)}{\left( \sum_{c=1}^{K} \exp(z_c) \right)^{\sum_c t_c}} \tag{7}$$

$$= \min -\frac{\exp(\sum_{c=1}^{K} t_c z_c)}{\sum_{c=1}^{K} \exp(z_c)} \tag{8}$$

$$= \min \underbrace{-\frac{\exp\left( \|\mathbf{w}\| \cdot \|\mathbf{f}\| \sum_{c=1}^{K} t_c \cos \theta_c + \sum_{c=1}^{K} t_c b_c \right)}{\sum_{c=1}^{K} \exp(\|\mathbf{w}\| \cdot \|\mathbf{f}\| \cos \theta_c + b_c)}}_{\triangleq L} \tag{9}$$

$$= \min L. \tag{10}$$

We use $\|\mathbf{w}_c\| = \|\mathbf{w}\|$ as assumed.

To solve this optimization problem, we compute the derivative of $L$ with respect to $\|\mathbf{f}\|$. For notational simplicity, let

$$A \triangleq \sum_c t_c z_c = \|\mathbf{w}\| \cdot \|\mathbf{f}\| \sum_c t_c \cos \theta_c + \sum_c t_c b_c, \qquad B \triangleq \sum_c \exp(z_c).$$

Then $L = -\exp(A)/B$. Since the bias $b_c$ does not depend on $\|\mathbf{f}\|$,

$$\frac{\partial z_c}{\partial \|\mathbf{f}\|} = \|\mathbf{w}\| \cos \theta_c.$$

Therefore,

$$\frac{\partial L}{\partial \|\mathbf{f}\|} = -\frac{\|\mathbf{w}\| \left( \sum_c t_c \cos \theta_c \right) \exp(A) \, B}{B^2} + \frac{\exp(A) \left( \|\mathbf{w}\| \sum_c \cos \theta_c \exp(z_c) \right)}{B^2} \tag{11}$$

$$= \underbrace{\frac{\|\mathbf{w}\| \exp(A)}{B^2}}_{\triangleq C} \times \underbrace{\left\{ -\left( \sum_c t_c \cos \theta_c \right) B + \sum_c \cos \theta_c \exp(z_c) \right\}}_{\triangleq D} \tag{12}$$

with $C > 0$.

$D$ can be rewritten as

$$D = \sum_c \exp(z_c) \left( \cos \theta_c - \sum_j t_j \cos \theta_j \right) \tag{13}$$

$$= B \left( \sum_c p_c \cos \theta_c - \sum_c t_c \cos \theta_c \right), \tag{14}$$

where we define $p_c \triangleq \frac{\exp(z_c)}{B}$.

Thus,

$$\frac{\partial L}{\partial \|\mathbf{f}\|} = C \, B \left( \sum_c p_c \cos \theta_c - \sum_c t_c \cos \theta_c \right). \tag{15}$$

**(i) Hard-label training.** For hard labels, *i.e.*, $t_y = 1$ and $t_{c \neq y} = 0$, Eq. 15 reduces to

$$\frac{\partial L_{\text{hard}}}{\partial \|\mathbf{f}\|} = C \, B \left( \sum_c p_c \cos \theta_c - \cos \theta_y \right), \tag{16}$$

where $C > 0$ and $B > 0$. Since the feature is assumed to be better aligned with the correct class, *i.e.* $\cos \theta_y > \cos \theta_{c \neq y}$, the weighted average $\sum_c p_c \cos \theta_c$ satisfies

$$\sum_c p_c \cos \theta_c \; < \; \cos \theta_y,$$

because $\{p_c\}$ is a probability distribution. Thus $\sum_c p_c \cos \theta_c - \cos \theta_y$ in eq. 16 is strictly negative, and hence

$$\frac{\partial L_{\text{hard}}}{\partial \|\mathbf{f}\|} < 0 \quad \text{for all finite } \|\mathbf{f}\|.$$

Therefore the loss strictly decreases as $\|\mathbf{f}\|$ increases, and the optimal solution is achieved when

$$\|\mathbf{f}_{\text{hard}}^*\| \; \longrightarrow \; \infty.$$

**(ii) Soft-label training.** Applying eq. 15 to soft labels yields

$$\frac{\partial L_{\text{soft}}}{\partial \|\mathbf{f}\|} = C \, B \left( \sum_c p_c \cos \theta_c - \sum_c t_c \cos \theta_c \right), \tag{17}$$

where $t_y < 1$ and $t_c > 0$ for at least one $c \neq y$. The cross-entropy is convex with respect to the logits, and each logit $z_c = \|\mathbf{w}\| \|\mathbf{f}\| \cos \theta_c + b_c$ is affine in $\|\mathbf{f}\|$. Thus $L_{\text{soft}}$ is a convex function with respect to $\|\mathbf{f}\|$.

**Behavior as $\|\mathbf{f}\| \to \infty$.** Since $\cos\theta_y > \cos\theta_{c\neq y}$, the logit gap $z_y - z_c$ grows unbounded, implying

$$p_y \to 1, \qquad p_{c\neq y} \to 0.$$

Therefore

$$\sum_c p_c \cos\theta_c \;\longrightarrow\; \cos\theta_y,$$

while for soft labels,

$$\sum_c t_c \cos\theta_c \;<\; \cos\theta_y.$$

Hence $\sum_c p_c \cos\theta_c - \sum_c t_c \cos\theta_c$ in eq. 17 becomes strictly positive for large $\|\mathbf{f}\|$, and so

$$\frac{\partial L_{\mathrm{soft}}}{\partial \|\mathbf{f}\|} > 0 \qquad \text{when } \|\mathbf{f}\| \text{ is sufficiently large.}$$

Thus the loss increases without bound as $\|\mathbf{f}\| \to \infty$.

**Existence of a finite optimum.** By convexity and the fact that $L_{\mathrm{soft}}(\mathbf{f}) \to +\infty$ as $\|\mathbf{f}\| \to \infty$, the loss must attain a global minimum where

$$\|\mathbf{f}^*_{\mathrm{soft}}\| < \infty.$$

If $\|\mathbf{f}^*_{\mathrm{soft}}\| > 0$, then the minimum occurs when

$$\sum_c (t_c - p_c^*) \cos\theta_c = 0, \tag{18}$$

where

$$p_c^* = \frac{\exp\!\big(\|\mathbf{w}\| \cdot \|\mathbf{f}^*_{\mathrm{soft}}\| \cos\theta_c + b_c\big)}{\sum_j \exp\!\big(\|\mathbf{w}\| \cdot \|\mathbf{f}^*_{\mathrm{soft}}\| \cos\theta_j + b_j\big)}. \tag{19}$$

Otherwise the minimum is attained at $\|\mathbf{f}^*_{\mathrm{soft}}\| = 0$, which is trivially finite.

Thus soft-label training yields a finite optimal feature norm, whereas hard-label training forces $\|\mathbf{f}^*_{\mathrm{hard}}\| \to \infty$. Consequently,

$$\|\mathbf{f}^*_{\mathrm{soft}}\| < \|\mathbf{f}^*_{\mathrm{hard}}\|.$$

## K  MORE RESULTS ON THE EFFECT OF REGULARIZATION

Figs. 14-18 present additional examples on how regularization affects feature distributions in the original representation space, across various models. As discussed in Section 4, regularization reduces the RMS of features (top row), leading to less confident predictions (bottom row). Furthermore, the overall cosine similarity between features and class centers increases with regularization (middle row). We show attack success rates for the stronger attack configurations (PGD$^\mathrm{s}$ and AutoAttack$^\mathrm{s}$) in Figs. 19-24.

In Tab. 8, we provide additional experimental results for MobileNetV2, EfficientNet-B1, and ViT-B/16 using pretrained weights from PyTorch. In the PyTorch framework, these pretrained weights can be accessed using the names `IMAGENET1K_V1`, `IMAGENET1K_V2`, and `IMAGENET1K_SWAG_LINEAR_V1`. For MobileNetV2 and EfficientNet-B1, the V2 weights result from stronger regularization compared to V1, whereas for ViT-B/16, the V1 weights are derived from stronger regularization compared to Swag Linear V1. Details on the regularization hyperparameters (using soft labels) used to train these weights can be found in Tab. 7.

Consistent with the findings in Section 4, applying stronger regularization reduces the RMS of the features and increases the cosine similarity between the features and class centers. Moreover, the models become more underconfident, and the attack success rates of both FGSM and PGD attacks decrease.

## L  FEATURE SCALING

In Section 4.3, we show that regularization using soft labels has an effect of scaling down of features. Here, we examine the possibility of manual feature scaling for calibration after training without regularization. In Fig. 25, we show the accuracy and calibration performance of ResNet50, Swin-T, MobileNetV2, and ConvNeXt-T on the ImageNet validation set, before and after manually scaling features across various models and weights. $S = 1$ (where $S$ indicates the scaling factor) represents the use of original features. It can be observed that manual feature scaling does not affect classification accuracy but can improve calibration due to its similarity to temperature scaling.

## M  EFFECT OF SOFT-LABEL REGULARIZATION ON DOWNSTREAM TRANSFER PERFORMANCE

To further investigate the broader implications of soft-label regularization beyond calibration and adversarial robustness, we evaluate its impact on downstream transfer learning. Specifically, we conduct linear-probe experiments on CIFAR-100, using features extracted from ImageNet-pretrained models. For each architecture, the feature extractor is frozen and a linear classifier is trained on CIFAR-100, following Li et al. (2022); Chen et al. (2020).

The results are summarized in Tab. 10. Across all architectures, we observe that soft-label regularization does not improve downstream linear-probe accuracy. In many cases, performance degrades when regularization is applied, despite the higher accuracy in the original ImageNet classification task.

Table 10: Linear-probe accuracy (%) on CIFAR-100 using ImageNet-pretrained features. Soft-label regularization does not improve downstream transfer performance and often degrades accuracy compared to the baseline.

| Method | ResNet-50 | MobileNetV2 | Swin-T | ConvNeXt-T |
|---|---|---|---|---|
| Baseline | 57.2 | 51.0 | 74.1 | 72.8 |
| Label Smoothing | 53.4 | 51.4 | 73.1 | 70.1 |
| Mixup | 55.0 | 50.3 | 74.9 | 75.0 |
| CutMix | 43.3 | 42.7 | 74.0 | - |

This observation aligns with recent findings showing that label smoothing increases ImageNet accuracy but reduces linear-probe transfer performance on CIFAR-10 (Zhou et al., 2025). Their analysis attributes this degradation to reduced transferability of features, as soft-label regularizers suppress informative, discriminative variations that are useful for downstream tasks. In particular, Zhou et al. (2025) report that soft-label supervision induces a form of representation collapse that harms performance on tasks requiring broad, expressive embeddings beyond the pretraining classes.

Our representation-space analysis provides a coherent explanation for this phenomenon. As shown in Section 4, soft-label regularization reduces the RMS of features and increases their alignment with class centers, producing tighter clusters. While such compact representations help calibration and adversarial robustness by reducing overconfident feature magnitudes and directing perturbation gradients toward the origin, they also shrink the diversity and expressiveness of the learned features. Downstream transfer tasks, however, rely on generalizable representations rather than class-specific clustering of features (Zhou et al., 2025). Thus, the observed degradation in linear-probe accuracy appears to be a natural consequence of the representation-space dynamics induced by soft-label regularization, yielding more calibrated and robust but less transferable representations.

## N  WHY ROBUSTNESS GAINS DIMINISH UNDER STRONG ITERATIVE ATTACKS

In Section 4.2, we show that regularization using soft labels enhances alignment between features and their corresponding class centers. As illustrated in Fig. 5, this enhanced alignment causes per-

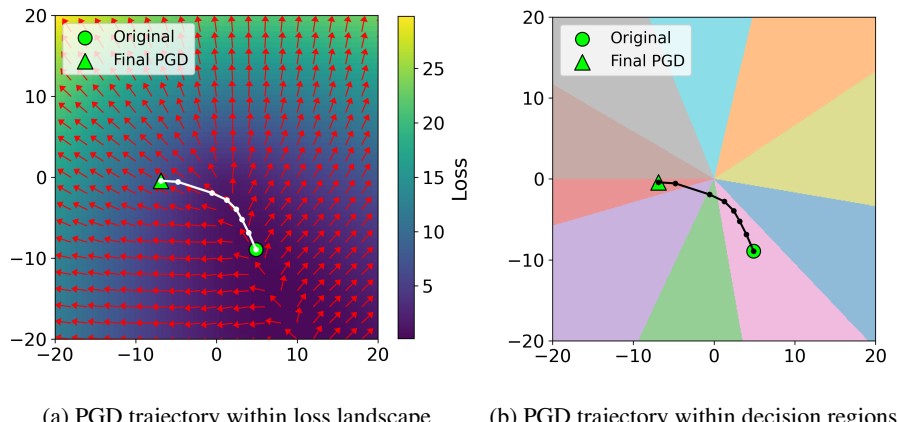

(a) PGD trajectory within loss landscape  (b) PGD trajectory within decision regions

Figure 13: PGD Trajectory of ResNet50 with label smoothing (7-step PGD, $\alpha = 2/255$, $\epsilon = 8/255$). The green circle denotes the representation of the original clean sample, and the green triangle indicates the representation after 7 PGD steps. The white/black markers between them correspond to intermediate representations at each PGD iteration. (a) shows the PGD trajectory within the loss landscape, where each PGD step follows the gradient direction at its current location. (b) shows the trajectory within the decision regions, illustrating that although the initial perturbation steps pull the feature toward the origin, the accumulated steps eventually push it across the decision boundary, resulting in misclassification.

turbations generated by gradient-based methods such as FGSM to move features toward the origin rather than toward nearby decision boundaries, thereby improving robustness to such attacks.

However, this advantage mainly holds during the early stages for stronger iterative attacks. As shown by the PGD trajectory in Fig. 13, the initial PGD steps perturb the feature toward the origin due to its improved alignment with the class center. As iterations proceed, however, the gradient directions begin to shift outward, causing later PGD steps to move the feature away from the origin. Once this outward drift accumulates sufficiently, the feature is eventually pushed across the decision boundary.

This phenomenon directly explains why our quantitative results in Tab. 2 show increased robustness under weaker attacks (*e.g.* FGSM, PGD[w], and AutoAttack[w]), whereas the robustness gains become much less evident under stronger settings (*e.g.* PGD[s] and AutoAttack[s]). For weaker settings, features with higher cosine similarity to their class centers remain robust because the perturbation steps move them inward in the representation space. Under stronger attacks, however, the accumulated perturbation eventually overwhelms this advantage, and the feature is ultimately pushed across the decision boundary, causing the robustness benefits of improved alignment to diminish. This explains why models show enhanced robustness to single-step and weak iterative attacks, yet provide limited protection when the attack strengths becomes sufficiently large.

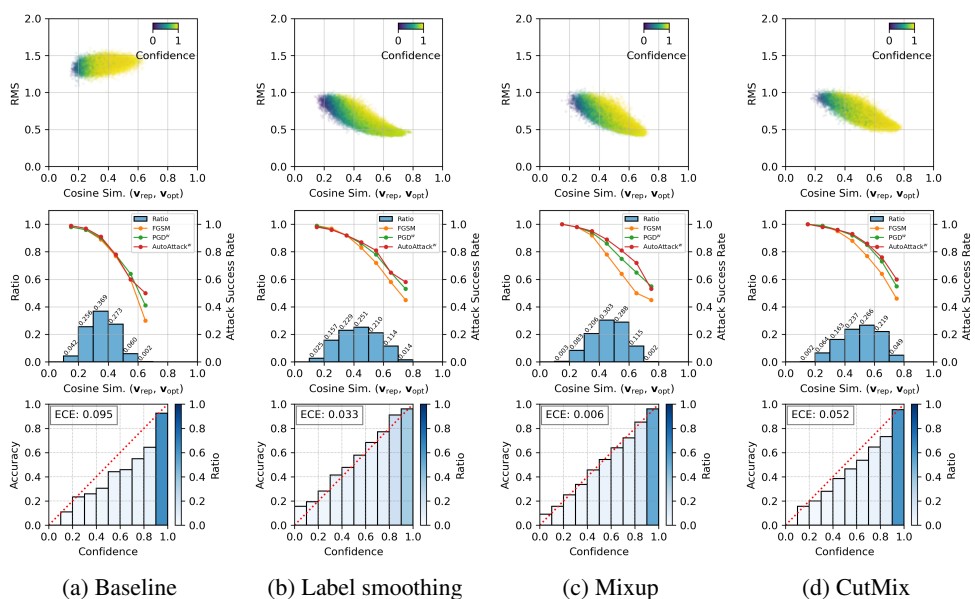

Figure 14: Evaluation results of Swin-T on the ImageNet validation data. **Top.** Scatter plots of feature RMS and cosine similarities of features ($v_{rep}$) with the class center ($v_{opt}$). Colors represent confidence values. **Middle.** Histograms of cosine similarities of features to class centers, along with the attack success rates of FGSM, PGD$^w$, and AutoAttack$^w$ for each bin (hyperparameters settings for the attacks can be found in Section 4.4). **Bottom.** Reliability diagrams, where the transparency of bars represents the ratio of data in each confidence bin. Expected calibration error (ECE) (Guo et al., 2017) values are shown for each case.

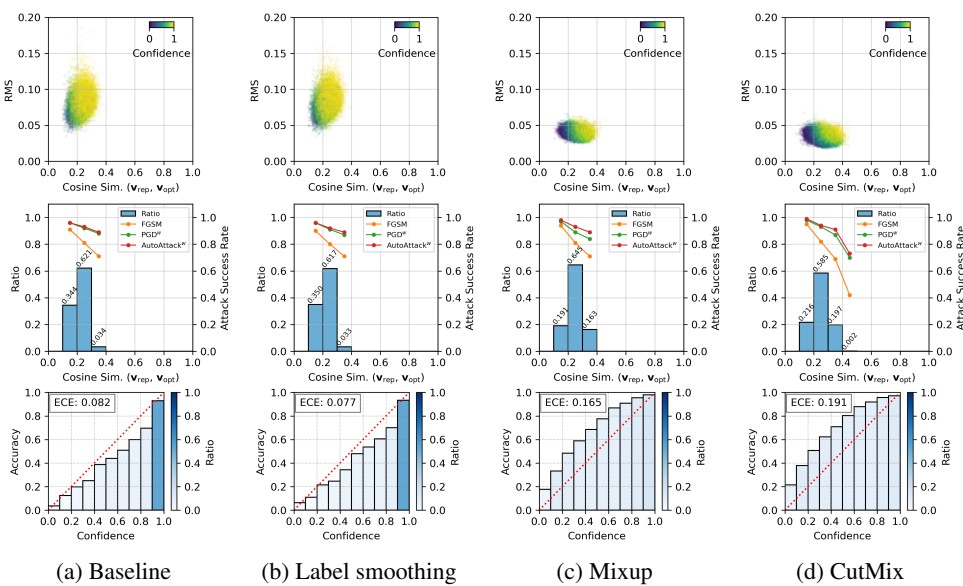

Figure 15: Evaluation results of MobileNetV2 on the ImageNet validation data. **Top.** Scatter plots of feature RMS and cosine similarities of features ($v_{rep}$) with the class center ($v_{opt}$). Colors represent confidence values. **Middle.** Histograms of cosine similarities of features to class centers, along with the attack success rates of FGSM, PGD$^w$, and AutoAttack$^w$ for each bin (hyperparameters settings for the attacks can be found in Section 4.4). **Bottom.** Reliability diagrams, where the transparency of bars represents the ratio of data in each confidence bin. Expected calibration error (ECE) (Guo et al., 2017) values are shown for each case.

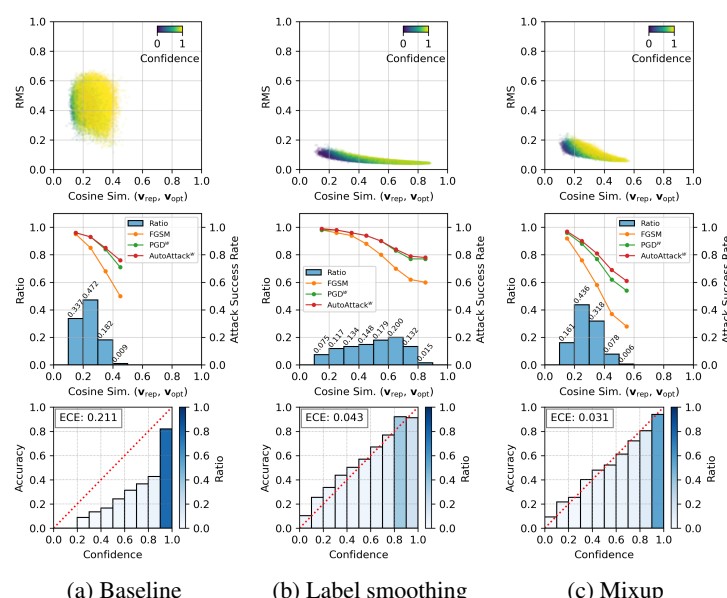

(a) Baseline      (b) Label smoothing      (c) Mixup

Figure 16: Evaluation results of ConvNeXt-T on the ImageNet validation data. **Top.** Scatter plots of feature RMS and cosine similarities of features ($\mathbf{v}_{rep}$) with the class center ($\mathbf{v}_{opt}$). Colors represent confidence values. **Middle.** Histograms of cosine similarities of features to class centers, along with the attack success rates of FGSM, PGD$^w$, and AutoAttack$^w$ for each bin (hyperparameters settings for the attacks can be found in Section 4.4). **Bottom.** Reliability diagrams, where the transparency of bars represents the ratio of data in each confidence bin. Expected calibration error (ECE) (Guo et al., 2017) values are shown for each case.

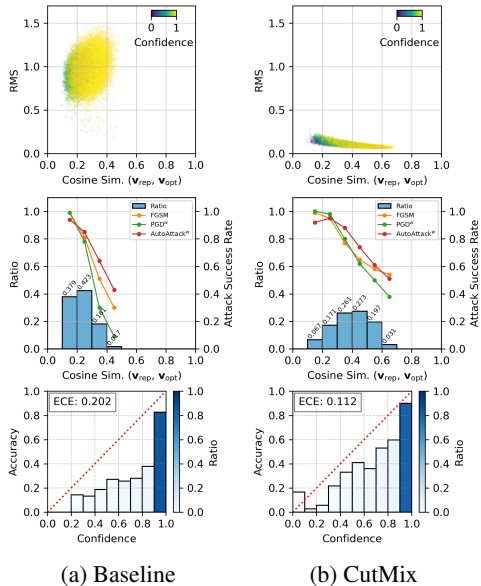

(a) Baseline      (b) CutMix

Figure 17: Evaluation results of ViT-B-16 on the ImageNet validation data. **Top.** Scatter plots of feature RMS and cosine similarities of features ($\mathbf{v}_{rep}$) with the class center ($\mathbf{v}_{opt}$). Colors represent confidence values. **Middle.** Histograms of cosine similarities of features to class centers, along with the attack success rates of FGSM, PGD$^w$, and AutoAttack$^w$ for each bin (hyperparameters settings for the attacks can be found in Section 4.4). **Bottom.** Reliability diagrams, where the transparency of bars represents the ratio of data in each confidence bin. Expected calibration error (ECE) (Guo et al., 2017) values are shown for each case.

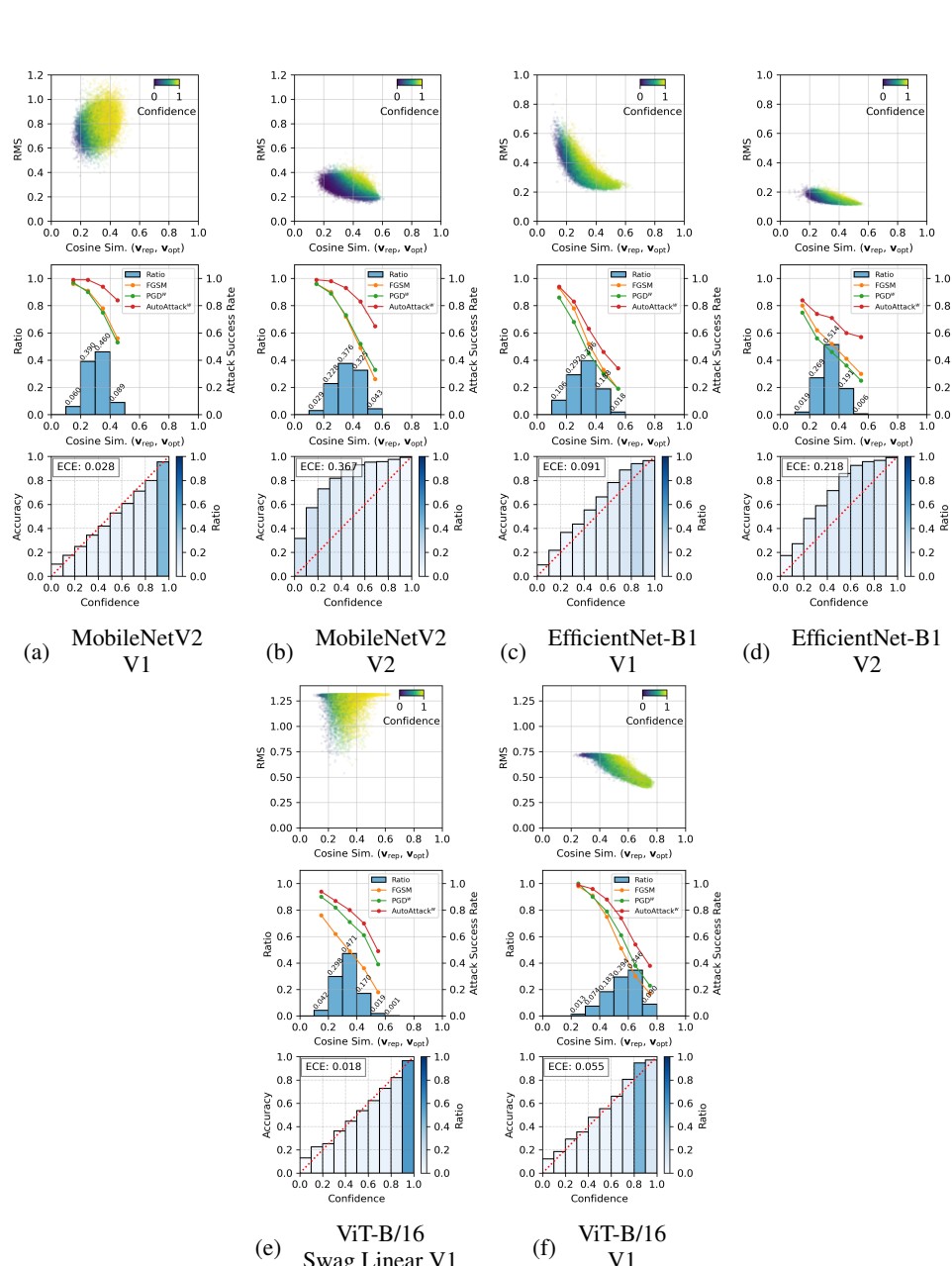

Figure 18: Evaluation results of MobileNetV2, EfficientNet-B1, and ViT-B/16 on the ImageNet validation data. **Top.** Scatter plots of feature RMS and cosine similarities of features ($\mathbf{v}_{rep}$) with the class center ($\mathbf{v}_{opt}$). Colors represent confidence values. **Middle.** Histograms of cosine similarities of features to class centers, along with the attack success rates of FGSM, PGD$^w$, and AutoAttack$^w$ for each bin (hyperparameters settings for the attacks can be found in Section 4.4). **Bottom.** Reliability diagrams, where the transparency of bars represents the ratio of data in each confidence bin. Expected calibration error (ECE) (Guo et al., 2017) values are shown for each case.

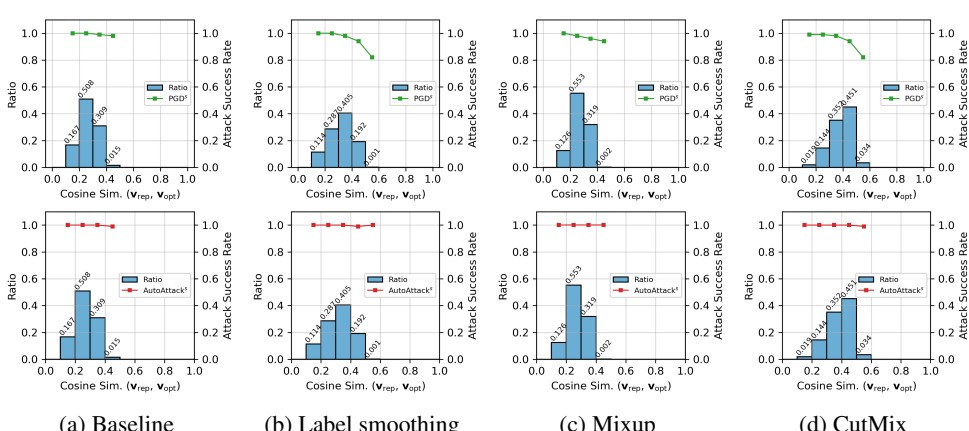

(a) Baseline    (b) Label smoothing    (c) Mixup    (d) CutMix

Figure 19: Evaluation results of ResNet50 on the ImageNet validation data. We show histograms of cosine similarities of features to class centers, along with the attack success rates for PGD$^s$ (top) and AutoAttack$^s$ (bottom) for each bin (hyperparameters settings for the attacks can be found in Section 4.4).

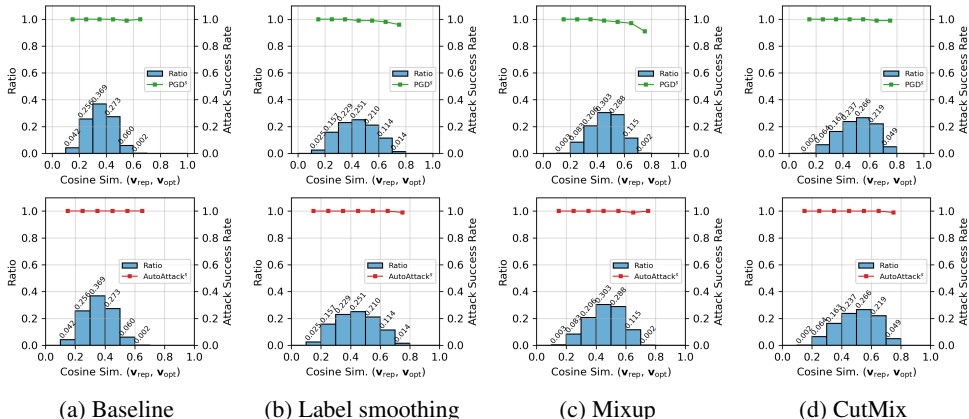

(a) Baseline    (b) Label smoothing    (c) Mixup    (d) CutMix

Figure 20: Evaluation results of Swin-T on the ImageNet validation data. We show histograms of cosine similarities of features to class centers, along with the attack success rates for PGD$^s$ (top) and AutoAttack$^s$ (bottom) for each bin (hyperparameters settings for the attacks can be found in Section 4.4).

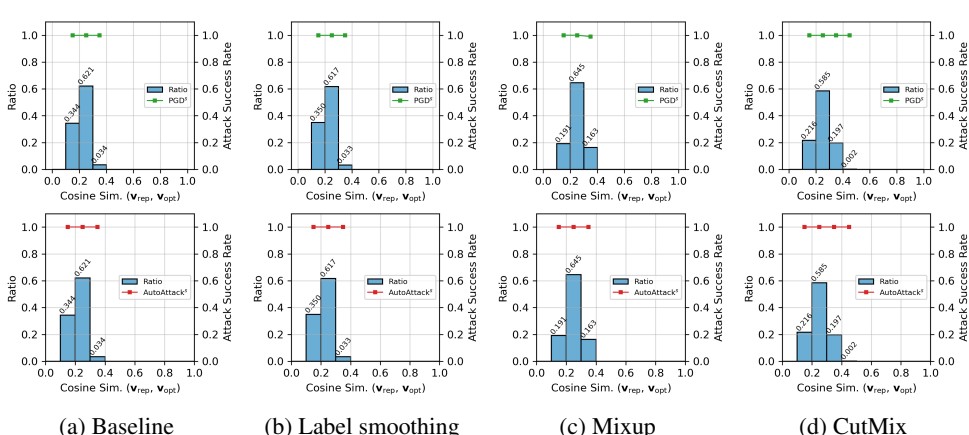

(a) Baseline      (b) Label smoothing      (c) Mixup      (d) CutMix

Figure 21: Evaluation results of MobileNetV2 on the ImageNet validation data. We show histograms of cosine similarities of features to class centers, along with the attack success rates for PGD[s] (top) and AutoAttack[s] (bottom) for each bin (hyperparameters settings for the attacks can be found in Section 4.4).

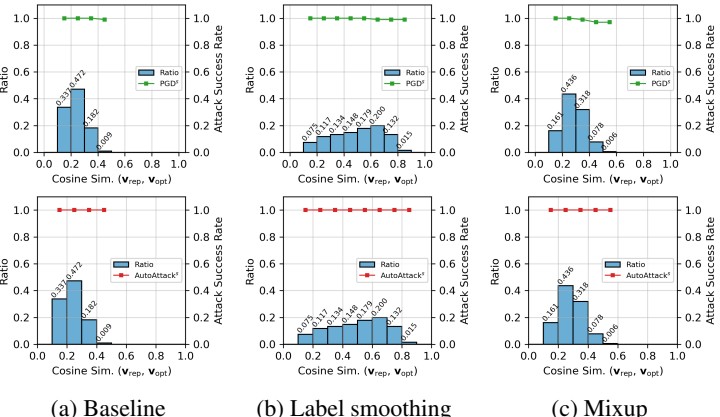

(a) Baseline      (b) Label smoothing      (c) Mixup

Figure 22: Evaluation results of ConvNeXt-T on the ImageNet validation data. We show histograms of cosine similarities of features to class centers, along with the attack success rates for PGD[s] (top) and AutoAttack[s] (bottom) for each bin (hyperparameters settings for the attacks can be found in Section 4.4).

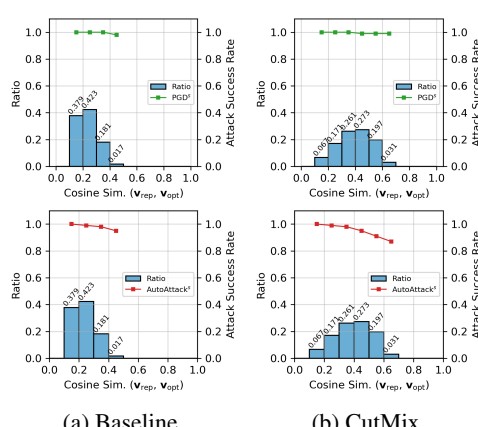

(a) Baseline  (b) CutMix

Figure 23: Evaluation results of ViT-B-16 on the ImageNet validation data. We show histograms of cosine similarities of features to class centers, along with the attack success rates for PGD[s] (top) and AutoAttack[s] (bottom) for each bin (hyperparameters settings for the attacks can be found in Section 4.4).

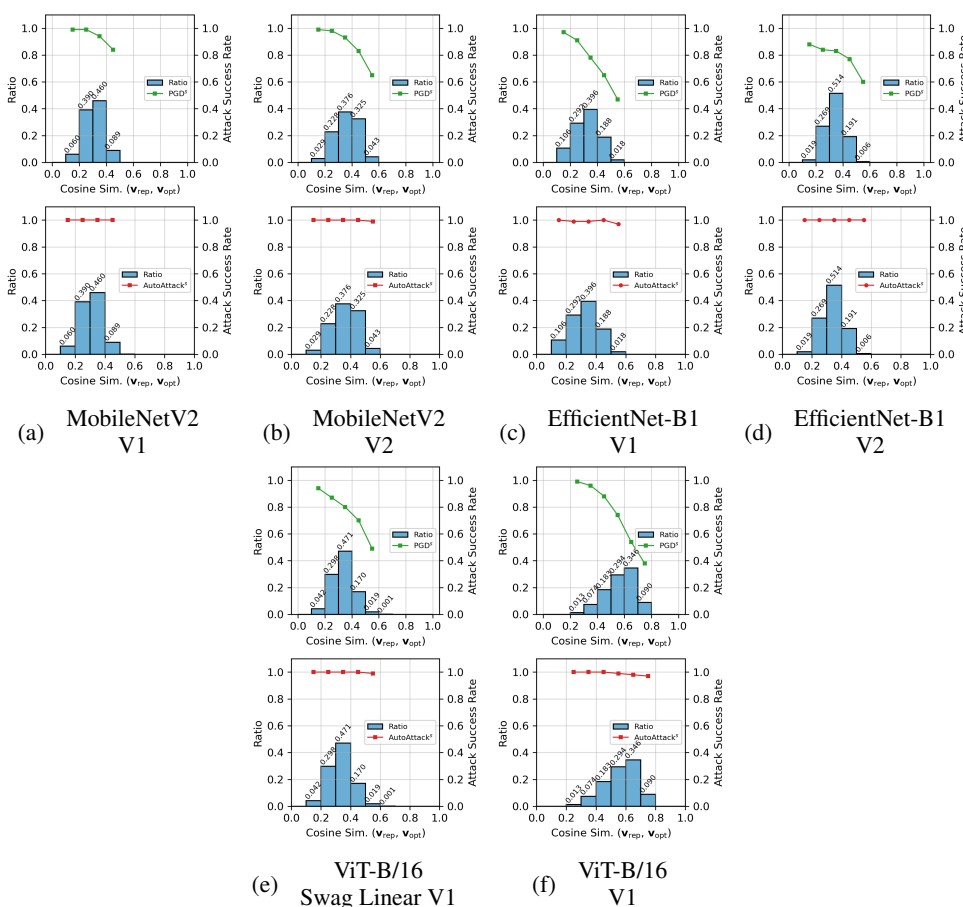

(a) MobileNetV2 V1  (b) MobileNetV2 V2  (c) EfficientNet-B1 V1  (d) EfficientNet-B1 V2

(e) ViT-B/16 Swag Linear V1  (f) ViT-B/16 V1

Figure 24: Evaluation results of MobileNetV2, EfficientNet-B1, and ViT-B/16 on the ImageNet validation data. We show histograms of cosine similarities of features to class centers, along with the attack success rates for PGD[s] (top) and AutoAttack[s] (bottom) for each bin (hyperparameters settings for the attacks can be found in Section 4.4).

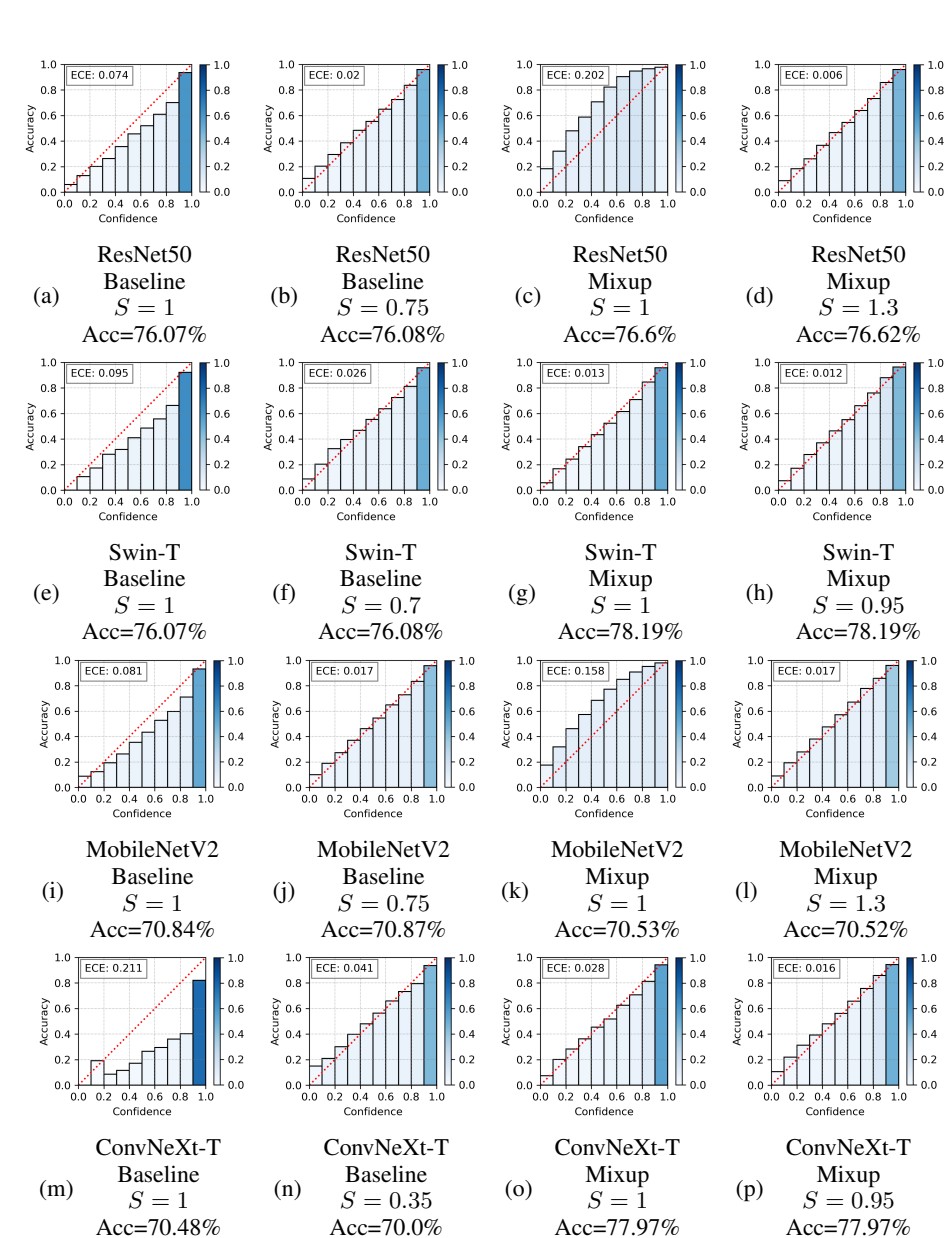

Figure 25: Calibration performances before and after manual feature scaling. $S$ represents the scaling factor, and Acc represents the accuracy on the ImageNet validation data.

