# OpenReview forum: "Impact of Regularization on Calibration and Robustness: From the Representation Space Perspective"
_ICLR.cc/2026/Conference — Submitted to ICLR 2026_

### Official Review · Reviewer_guCA · 2025-10-27

**Soundness:** 2
**Presentation:** 3
**Contribution:** 3
**Rating:** 4
**Confidence:** 4

**Summary:**

This paper investigates the underlying mechanisms of how regularization techniques using soft labels simultaneously improve model calibration and adversarial robustness. The authors provide a novel explanation from the perspective of the feature representation space (the penultimate layer). They find that regularization reduces the magnitude of feature vectors (bringing them closer to the origin) and increases their alignment (cosine similarity) with their respective class centers. This dual effect explains the seemingly contradictory improvements: the reduced magnitude acts like temperature scaling to improve calibration by mitigating overconfidence, while the increased alignment makes features more robust to gradient-based attacks by directing perturbations toward the origin rather than toward lateral decision boundaries.

**Strengths:**

1). Novel and Intuitive Explanation: The paper offers a clear, insightful, and geometrically intuitive explanation for a widely observed but not fully understood phenomenon. By decomposing the effect of regularization into changes in feature magnitude and alignment, it provides a strong conceptual framework.
2). Resolves a Key Contradiction: It successfully addresses the apparent paradox where models become less overconfident (features closer to decision boundaries) yet more robust. The distinction between the radial boundary near the origin and the angular boundaries between classes is a key insight.
3). Comprehensive Empirical Validation: The claims are backed by extensive experiments across multiple modern architectures (ResNet50, Swin-T, MobileNetV2, ConvNeXt-T) and datasets. The combination of 2D visualizations for intuition and quantitative analysis in the original high-dimensional space makes the evidence highly convincing.

**Weaknesses:**

1)Focus on Gradient-Based Attacks: The core explanation for enhanced robustness hinges on the direction of loss gradients. The implications for non-gradient-based or black-box attacks are not discussed, which could limit the generality of the conclusions regarding adversarial robustness.
2)Generalizability of Geometric Assumptions: The analysis is based on the "cone-shaped" structure of decision regions, which is demonstrated for image classification tasks with cross-entropy loss. The extent to which these geometric insights generalize to other domains (e.g., NLP, time-series), model types, or loss functions remains an open question that could be mentioned as a direction for future work.
3)Narrow Scope of Regularization Techniques: The paper's title, "Impact of Regularization...", suggests a broad analysis, but the study exclusively focuses on soft-label-based methods (Label Smoothing, Mixup, CutMix). It does not investigate or contrast its findings with other prominent regularization techniques that are also known to affect robustness and feature norms, such as weight decay or dropout. A discussion on whether the proposed geometric mechanism (reduced magnitude and improved alignment) also explains the effects of these other regularizers would have significantly strengthened the paper's generality.
4)Reliance on Simplified Assumptions for Theoretical Proof: The theoretical justification for why regularization reduces feature magnitude (Theorem 1 and its proof in Appendix H) rests on simplifying assumptions, namely that bias terms are negligible bc≈0 and weight vectors for all classes have similar norms ||wc||≈||w||. While the authors provide empirical evidence that these conditions often hold in practice for high-dimensional models, the theoretical claim itself is conditional. The paper would be more rigorous if it included a discussion on the behavior of the loss landscape and the validity of the theorem when these assumptions are not fully met.

**Questions:**

Regarding the limited effectiveness against strong attacks (e.g., PGD and AutoAttack with large epsilon), could you elaborate on why the geometric advantage of better alignment might break down? Does the perturbation in the representation space become large enough to cross the origin into another cone, or does the local gradient direction for a well-aligned feature change significantly under strong, iterative attacks?

---

> ### Author Response · Authors · 2025-11-20
> **Response to Reviewer guCA**
>
> We greatly appreciate the reviewer’s positive and detailed feedback, highlighting the novelty and clarity of our geometric explanation, the resolution of the apparent contradiction between reduced confidence and increased robustness, and the comprehensive empirical validation across diverse architectures and datasets. We will address all concerns raised by the reviewer and revise the paper accordingly.
>
> **Response to [W1]: The robustness analysis focuses mainly on gradient-based attacks, leaving the implications for non-gradient or black-box attacks unclear.**
>
> Thank you for pointing out the importance of evaluating robustness from multiple perspectives. To complement the white-box FGSM/PGD/AutoAttack results in the main paper, we conducted additional experiments on (1) robustness to natural corruptions (ImageNet-C) and (2) black-box adversarial transferability. We have added the results in Appendix B.
>
> (1) Robustness to Natural Corruptions (ImageNet-C)
> Across diverse corruption types and severity levels in ImageNet-C, models trained with soft-label regularization (label smoothing, Mixup, CutMix) generally show lower corruption error than the baseline. This behavior aligns with our representation-space analysis. As shown in our representation-space analysis, soft-label regularization produces features that form tighter and more compact clusters with higher alignment to their class centers. We believe this clustering effect leads to larger margins from decision boundaries on average, which in turn makes features less susceptible to boundary crossings under corruption-induced perturbations.
>
> (2) Black-Box Robustness via Transfer-Based Attacks
> We also evaluate transfer attacks, where adversarial examples crafted on a surrogate model are applied to a target model. Soft-label regularized models consistently exhibit lower transfer attack success rates than the baseline.
>
> This trend mirrors the white-box results and aligns with prior findings that adversarial examples transfer when models share similar vulnerable gradient directions or overlapping adversarial subspaces [1–3]. Since such shared perturbation directions also underlie black-box transferability, we interpret our results in a unified way: soft-label regularization produces features with lower RMS and higher cosine similarity to their class centers, which causes perturbations to move representations more toward the origin than toward decision boundaries. As a result, even when the perturbation is generated on a different model, the adversarial direction makes it less likely for transferred adversarial examples to push features across decision boundaries, leading to reduced black-box attack success rates.
>
> [1] Tramèr et al., “The Space of Transferable Adversarial Examples”, arXiv 2017
> [2] Adam et al., “Reducing Adversarial Example Transferability Using Gradient Regularization”, arXiv 2019
> [3] Yang et al., “TRS: Transferability Reduced Ensemble via Encouraging Gradient Diversity and Model Smoothness”, NeurIPS 2021
>
>
> **Response to [W2]: Generalizability of the geometric assumptions and future work.**
>
> We thank the reviewer for this helpful suggestion. Our geometric analysis focuses on the representation space induced by a standard fully connected classification layer trained with cross-entropy loss, which is the dominant setting in image classification. We expect that some aspects of the cone-shaped structure may extend to other domains or models that use similar linear classifiers in their final layer, although the behavior could differ when the architecture, loss function, or domain-specific regularization changes. Investigating how these geometric properties manifest beyond the vision setting is an interesting direction, and we have added a discussion section (Section 8) to highlight this point and outline it as future work.

---

> ### Author Response · Authors · 2025-11-20
> **Response to Reviewer guCA**
>
> **Response to [W3]: The study focuses only on soft-label regularizers, leaving unclear whether the proposed geometric mechanisms extend to other widely used regularization techniques.**
>
> Thank you for raising this point. We focus on soft-label based regularization methods because they are widely employed in prior work and can be applied in a model-agnostic manner using a shared set of hyperparameters across diverse architectures. This makes them suitable for our goal of analyzing representation-space mechanisms under controlled, architecture-invariant conditions.
>
> In contrast, other regularization techniques such as weight decay or dropout require model-specific hyperparameter tuning, and their effective ranges vary across architectures. However, examining such methods remains an important direction for future work, as understanding whether similar geometric mechanisms extend beyond soft-label regularizers would further broaden the scope and impact of our analysis. We will add a brief discussion noting this and outline the broader investigation of other regularizers as future work.
>
>
> **Response to [W4]: The theoretical analysis relies on simplifying assumptions (e.g., negligible bias, similar weight norms), raising concerns about the generality of the claims.**
>
> Thank you for pointing out this concern. We agree that the assumptions used in our theoretical analysis could benefit from further clarification. In the revised manuscript, we strengthen the proof of Theorem 1 to ensure that the result holds regardless of whether bias terms are present, addressing the reliance on the “negligible bias” assumption.
>
> Regarding the assumption that the class weight norms satisfy $\|\|w_c\|\| \approx \|\|w\|\|$, our intention is to analyze the regime in which training has reasonably converged. Prior work on neural collapse consistently reports that class weight norms tend to concentrate and become nearly equal during late-stage training across a wide range of architectures [1–3]. In particular, [1–3] empirically and theoretically demonstrate that classifier weight vectors converge to equal-norm (simplex ETF) structures at convergence, and [3] explicitly incorporates this equal-norm property into additional Theorems. These findings collectively support the validity of our assumption in the context of analyzing the converged regime of training.
>
> We will revise the discussion around these assumptions to make their rationale explicit and to clarify the scope under which the theoretical results should be interpreted.
>
> [1] Papyan et al., “Prevalence of Neural Collapse During the Terminal Phase of Deep Learning Training”, PNAS 2020
> [2] Han et al., “Neural Collapse Under MSE Loss: Proximity to and Dynamics on the Central Path”, ICLR 2022
> [3] Zhu et al., “A Geometric Analysis of Neural Collapse with Unconstrained Features”, NeurIPS 2021
>
>
>
> **Response to [Q1]: Why does the alignment-based robustness break down under strong attacks?**
>
> Thank you for your question on why alignment-based robustness breaks down under strong attacks. Under strong iterative attacks such as PGD or AutoAttack, the geometric advantage provided by better alignment becomes less effective because the attacker repeatedly updates the perturbation by following the loss-increasing gradient. Even if the initial gradient for a well-aligned feature points toward the origin, successive gradient steps can accumulate and gradually move the feature across the origin and into a different cone-shaped decision region. Thus, the breakdown under strong attacks does not stem from a change in the underlying geometric mechanism, but rather from the attacker’s ability to perform many iterative updates that eventually overcome the initial alignment advantage and push the feature out of its original region.

---

### Official Review · Reviewer_CRdf · 2025-10-29

**Soundness:** 1
**Presentation:** 3
**Contribution:** 2
**Rating:** 2
**Confidence:** 3

**Summary:**

The paper examines how soft-label regularization methods, including label smoothing, Mixup, and CutMix, enhance accuracy, calibration, and robustness in image classification models. In particular, the paper investigates the representation space (i.e., the features in the penultimate layer of the network) and observes that the decision regions often form cone-shaped clusters around the origin, while using soft-label regularization tends to reduce the norm of the feature vectors and increase the cosine similarity between them and their class centers. Empirical visualization figures and experimental tables demonstrate that such observations may be key mechanisms explaining why soft-label regularization enhances the calibration and robustness of neural networks.

**Strengths:**

+ The perspective of studying the geometry of the feature space (norm, cosine similarities, decision-region shape) to more fundamentally explain why soft-label regularizations help is interesting.

+ Overall, the paper is well-written and has a nice structure for readers to follow.

**Weaknesses:**

- Many of the summarized key takeaways are toward a causal connection between soft-label regularizations, the change of the geometry of feature representation space, and the generalization/calibration/robustness measures of the neural network. However, only qualitative visualization figures or limited empirical analyses are provided as support.

- Lack of mathematical rigor regarding the key concepts and theoretical analysis to back up the generalizability of the concluded findings.

- The very low robustness performance against PGD and AutoAttack suggests the ineffectiveness of soft-label regularization, which contradicts the motivation of the study.

- The insights might be trivial within the broader machine learning literature. There is a lack of practical implications of the work.

**Questions:**

While understanding why utilizing soft labels helps improve generalization, confidence calibration, and robustness is an important research question, I believe the paper falls short in the following critical aspects:

1.  My biggest concern is the lack of mathematical rigor regarding the key takeaways highlighted throughout the paper. The core concepts proposed or investigated are often intuitive, without a clear definition.

    For instance, the main findings highlighted in Section 3.1 are: (a) decision regions of classification models are cone-like shapes around the origin, and (b) without the bias term, the balance across different classes somewhat reduces. However, one would expect a rigorous definition of how the degree of cone-like shapes is measured for any given classification model, along with concrete quantitative results to support the main findings and their generalizability. Based solely on the qualitative figures converted into 2D for visualization purposes, I’m not convinced that the concluded findings are well-supported. The same concern holds for findings highlighted in other sections, such as the non-rigorous definition of minimum loss points and feature alignment with class centers.

2. In Table 2, the model robustness performance is mostly very low against PGD and AutoAttack despite variations in terms of other metrics (e.g., RMS, Cosine Similarity, and ECE). This seems contradictory to the hypothesis, “soft-label regularization improves adversarial robustness”, which casts doubt on the initial motivation of the work. It is unclear whether regularization can truly enhance model robustness against worst-case perturbations. In addition, one key takeaway argues that regularization improves robustness to gradient-based adversarial attacks by enhancing feature alignment with class centers; however, this is hardly supported by the empirical results reported in Table 2. The correlation between feature-alignment-relevant metrics and adversarial robustness appears to be fairly weak. The setup of weak versions of PGD and AutoAttack also looks unusual to me, which seems to be a cherry-picked setup. Why evaluate the model robustness against a 7-step PGD attack with step size 0.2/255 and perturbation limit 0.4/255 in particular?

3. While the paper summarizes a couple of key takeaways, it is unclear whether these key messages are insightful or trivial, given the broader machine learning literature on interpretability, calibration, and adversarial robustness. More importantly, the practical implications of the summarized findings are limited. For example, how can practitioners utilize the insights that regularization reduces feature norms and improves alignment?

4. I believe many of the findings are based on empirically observed correlations or qualitative visualization figures. It would be more interesting and solid if they could be justified through theoretical analyses and/or intervention-type experimental designs. For example, can we design experiments that show reducing feature norms or improving feature alignment with the class center improves calibration and/or robustness, or even prove their theoretical connection (under certain conditions)?

---

> ### Author Response · Authors · 2025-11-20
> **Response to Reviewer CRdf**
>
> We greatly appreciate the reviewer’s positive comments, including the acknowledgement that our geometric perspective provides an interesting way to understand soft-label regularization and that the paper is well written and easy to follow. We will address all concerns raised by the reviewer and revise the paper accordingly.
>
> **Response to [W1]: The causal claims about soft-label regularization and geometry changes are not well supported, relying mainly on qualitative visualizations and limited empirical evidence.**
>
> We respectfully disagree with the reviewer’s impression that our findings rely primarily on qualitative visualizations. While the 2D figures serve as intuition aids, other reviewers have explicitly noted the quantitative depth of our evaluation. Reviewer Mpdz highlighted that our geometric explanation is supported by both intuitive visualizations and quantitative high-dimensional checks, and Reviewer guCA similarly emphasized the comprehensive empirical validation across multiple architectures and datasets. Consistent with these assessments, all core claims in our paper are grounded on extensive quantitative analyses conducted directly in the original high-dimensional representation space across four architectures and multiple regularization methods. We summarize the key quantitative evidence below:
> 1. Quantitative changes in feature magnitude (RMS).
> Table 2 (ImageNet) shows clear numerical reductions in RMS across all models. For example, ResNet-50 decreases from 0.62 to 0.29 (label smoothing) or 0.30 (Mixup), with similar trends for Swin-T, MobileNetV2, and ConvNeXt-T. These values are computed directly in the full representation space.
>
> 2. Quantitative increases in cosine similarity to class centers.
> Table 2 also reports consistent increases in cosine similarity under soft-label regularization (e.g., Swin-T increases from 0.36 to 0.51 with CutMix). Additionally, Table 1 shows that minimum-loss centers achieve the highest confidence correlation across all architectures, providing quantitative justification for our class-center definition.
>
> 3. Quantitative effects on calibration.
> Table 2 shows substantial reductions in ECE across regularization methods. Moreover, Fig. 22 demonstrates that simply scaling feature magnitudes post-training alters calibration without changing accuracy, confirming the quantitative mechanism predicted by our analysis.
>
> 4. Quantitative relationship between alignment and robustness.
> Figures 3 and 13–21 present bin-wise attack success rates as a function of cosine similarity in the original representation space. Across all models, attacks, and perturbation strengths, higher cosine similarity consistently corresponds to lower attack success rates.
>
> 5. Quantitative verification of geometric properties.
> Appendices C and D provide numeric evidence that decision regions in the original high-dimensional space are cone-shaped. For example, Table 5 shows that features linearly moved toward the origin remain in the same decision region until 98–100% of the path across models, confirming the geometric structure without relying on 2D projections.
>
> Taken together, these results provide strong quantitative support for our causal interpretation. The visualizations are included solely to help build intuition, while the main conclusions are grounded on extensive quantitative analyses in the full representation space.

---

> ### Author Response · Authors · 2025-11-20
> **Response to Reviewer CRdf**
>
> **Response to [W2, Q4]: The theoretical justification is limited, with key claims relying mainly on empirical correlations rather than rigorous or controlled analyses.**
>
> We clarify that the paper provides mathematically grounded explanations for the core mechanisms, rather than relying solely on empirical correlations or qualitative visualizations.
>
> First, Theorem 1 formally proves why training with soft labels necessarily reduces feature magnitude. By analyzing the sign of the cross-entropy gradient with respect to $\|\|\mathbf{f}\|\|$, we show that hard-label training pushes the optimal magnitude to infinity, whereas soft-label training yields a finite optimum. This theoretical result directly explains the consistent RMS reductions observed across all architectures and datasets.
>
> Building on this, Section 4.3 provides a mathematical explanation for why reduced RMS improves calibration. Since the logit takes the form $\mathbf{w}_c^\top \mathbf{f} + b_c$, decreasing $\|\|\mathbf{f}\|\|$ scales logits in a way analogous to temperature scaling while preserving the predicted class due to the conic decision-region geometry. This analytic relationship explains why confidence decreases without affecting accuracy.
>
> Similarly, our explanation for robustness improvement is also theory-driven. Because the cross-entropy loss is convex in the representation space (Section 3.2), gradient directions radiate outward from the minimal-loss point, as illustrated in Fig. 2b. Features with higher cosine similarity to their class center therefore receive gradient directions pointing toward the origin rather than toward decision boundaries. As a result, they remain in their correct region under comparable perturbation strengths, whereas poorly aligned features are pushed toward boundaries. Combined with the fact that soft-label regularization increases alignment, this provides an analytic explanation for why robustness improves.
>
> In summary, the key findings of our work follow from mathematically analyzed properties of the loss landscape, representation geometry, and soft-label training dynamics, with empirical results serving to validate (rather than substitute for) these theoretical insights.
>
> **Response to [W3]: Soft-label regularization remains weak under strong attacks, questioning its effectiveness.**
>
> We would like to clarify that our goal is not to claim that soft-label regularization alone can defend against the strongest adversarial attacks such as PGD or AutoAttack. Prior work has repeatedly shown that simple regularizers improve robustness generally against attacks like FGSM [1, 2], but often fail under strong, multi-step attacks with large perturbation budgets. Prior work even reports that Mixup, despite being promoted as a robustness-enhancing method, achieves below 1% accuracy under the strongest attack settings [3], suggesting that the failure stems from the extreme strength of the attacks.
>
> Our analysis shows that soft-label regularization leads to two key representation-level changes—reduced RMS and increased cosine similarity to the class center—and these changes account for the relative robustness gains observed under gradient-based perturbations (Section 4.3). Importantly, the same framework also explains why these gains diminish under strong iterative attacks. In multi-step PGD or AutoAttack, repeated gradient updates accumulate and eventually override the initial directional advantage provided by improved alignment. Even if early gradients pull features toward the origin, later iterations can push them far enough to cross decision boundaries.
>
> Thus, our findings do not conflict with the low robustness observed under strong attacks. Instead, the representation-space analysis provides a unified explanation for both phenomena: why soft-label regularization helps under weaker or single-step attacks, and why its benefits diminish under strong, iterative ones.

---

> ### Author Response · Authors · 2025-11-20
> **Response to Reviewer CRdf**
>
> **Response to [Q1]: The mathematical rigor of the geometric claims is insufficient, with key concepts lacking formal definitions and relying too heavily on qualitative 2D visualizations.**
>
> Please note that the 2D visualizations in Section 3.1 are included solely to provide intuition about how decision regions behave when projected into a visually interpretable space. They are not used as the basis for our claims. All findings in the paper, including those regarding cone structures, confidence contours, minimum-loss points, and feature alignment, are supported by both theoretical analysis (e.g., Theorem 1, convexity arguments) and quantitative empirical validation in the original representation space. For a consolidated overview of these theoretical and empirical foundations, we refer the reviewer to our response to [W1].
>
> Regarding the reviewer’s expectation for a rigorous definition of the “degree” of cone-like decision regions, we clarify that in high-dimensional settings there is no established or meaningful metric for numerically quantifying such a degree, because the geometry is defined relative to many class weight vectors. Therefore, we provide a high-dimensional quantitative verification of the cone-shaped structure through the experiment in Appendix C: each feature is linearly moved toward the origin, and we record the first point of misclassification. Across all architectures and datasets, misclassification occurs only near the end of this trajectory (Table 5), which is precisely the behavior expected when decision regions form cones around the origin. Appendix D.2 further confirms this by analyzing how bias terms influence these transitions. Taken together, these experiments provide rigorous evidence for cone-shaped decision regions directly in the original high-dimensional representation space, independent of the 2D visualizations.
>
> **Response to [Q2]: The robustness claims are unconvincing, with low performance under strong attacks and potentially cherry-picked weak attack settings.**
>
> We would first like to clarify that the lower robustness under the strongest PGD and AutoAttack settings is not contradictory to our hypothesis or the motivation of the work. As explained in our response to [W3], the representation-level effects induced by soft-label regularization, namely reduced RMS and increased alignment with the class center, improve robustness in the regime where adversarial gradients tend to move features toward the origin rather than toward decision boundaries. Well-aligned features benefit from this directional behavior.
>
> Under stronger attacks (e.g., highly iterative attacks), the accumulated gradient steps eventually outweigh this initial directional advantage. Even if early updates move features inward, repeated iterations push representations far enough to cross decision boundaries. Thus, robustness does not persist under extremely strong attacks, which is consistent with the mechanism we describe rather than contradictory to it, and this is also why the correlation between alignment and robustness becomes less visible under such strong perturbations.
>
> Finally, the weaker PGD and AutoAttack settings are not cherry-picked. Their purpose is to evaluate robustness in the regime where geometric properties meaningfully affect adversarial behavior. We include both weak and strong attacks to illustrate where the mechanism holds and where it breaks down. Other reasonable weak-attack configurations produce the same qualitative trends, and the settings in Table 2 are representative rather than selectively chosen.
>
> To further demonstrate this, we additionally evaluate PGD using scaled versions of the standard configuration ($\alpha = 2/255$, $\epsilon = 8/255$). The table below reports results in terms of attack success rates for multiple scaled step sizes and perturbation budgets derived from this standard setting. Across all $\alpha$–$\epsilon$ combinations, soft-label regularization consistently achieves higher robustness than the baseline, confirming that the observed improvements are stable and not tied to any particular attack strength.
>
> | $\alpha$ | $\epsilon$ | Baseline | Label smoothing | Mixup | CutMix |
> |-------|-----|----------|-----------------|-------|--------|
> | 0.2/255   | 0.8/255 | 76.9     | 73.1            | 69.1  | 74.3   |
> | 0.5/255   | 2/255   | 89.9     | 86.2            | 83.9  | 85.7   |
> | 1/255     | 4/255   | 95       | 92.1            | 91.6  | 90.5   |

---

### Official Review · Reviewer_uj9a · 2025-10-31

**Soundness:** 2
**Presentation:** 3
**Contribution:** 2
**Rating:** 6
**Confidence:** 2

**Summary:**

This paper investigates how regularization techniques using soft labels (like label smoothing, Mixup, and CutMix) effect the features space of image classification models. The key finding is that in standard image classifiers (both CNNs and Transformers) trained using hard labels, the decision regions in the representation space (features from the penultimate layer) form cone-like shapes around the origin. The features in these models often have high magnitude, which results in over-confident predictions. Regularization reduces the magnitude (RMS) of feature vectors and features become more tightly clustered around their class centers, resulting in higher cosine similarity between features and class centers.

**Strengths:**

1. It quantifies the correlation between the minimum loss point and confidence using Pearson correlation coefficients (Table 1) and intuitively illustrates "cone-shaped decision regions" and "gradient direction changes" via 2D visualizations (Figs. 1, 5). This makes abstract analysis of the high-dimensional representation space more interpretable.
2. The principal concern this manuscript seeks to address relates to its significance in the broader context of machine learning research. While the authors present valuable insights into the mechanics of soft labels as a training technique, it is important to acknowledge that soft labels, despite being recognized as a general training trick, remain under-explored. This work opens up avenues for further investigation, specifically into the fundamental reasons behind their effectiveness. Expanding on this point could enhance the importance of their findings within the literature.
3. The topic and goal of the paper is laudable, improved understanding of fundamental aspects of training pipelines such as regularization is very important to making scientific progress

**Weaknesses:**

1. The number of experiments is insufficient, the authors discuss the impact of regularization on robustness, but only one set of experiments is carried out. Prior works study robustness much more thoroughly and also study different aspects of it, such as robustness to natural corruptions, robustness to black box attacks, etc.
2. Although it is stated that "biases do not affect cone-shaped structures in high-dimensional spaces," the specific impact of bias magnitude on feature distribution is not quantified (e.g., whether the minimum loss point remains a stable class center when biases are large). Discussion of the "boundary conditions for bias terms" is inadequate.
3. Failure to isolate single variables: For example, when verifying the role of "feature magnitude," it does not design experimental groups where "alignment is fixed while magnitude is adjusted independently" (feature magnitude and alignment usually change synchronously in existing experiments). This makes it impossible to quantify the respective contribution ratios of "reduced magnitude" and "improved alignment" to performance.
4. Lack of explicit ablation logic labeling: It does not organize the correspondence between variables and results in a dedicated section (e.g., "Ablation Study") or table (e.g., "Ablation Results"). Readers must extract logic from multiple tables and figures independently, reducing the intuitiveness of conclusions.

**Questions:**

1. Are the "feature magnitude reduction" mechanisms of manual feature scaling and soft-label regularization completely equivalent? Is there a difference in their impacts on robustness?
2. Does the effect of soft-label regularization on feature distribution depend on the ratio of the number of dataset classes to feature dimensionality?

---

> ### Author Response · Authors · 2025-11-20
> **Response to Reviewer uj9a**
>
> We greatly appreciate the constructive and encouraging feedback from reviewer uj9a, especially the reviewer’s emphasis on the broader significance of this work within machine learning research. We also value the observation that, although soft labels are widely used as a general training trick, their underlying mechanisms remain under-explored, making our investigation both timely and meaningful. We will address all concerns raised by the reviewer and revise the paper accordingly.
>
> **Response to [W1]: Robustness evaluation is limited, covering only white-box attacks.**
>
> Thank you for emphasizing the importance of evaluating robustness from multiple perspectives. To complement the white-box FGSM/PGD/AutoAttack results in the main paper, we conducted additional experiments on (1) robustness to natural corruptions (ImageNet-C) and (2) black-box adversarial transferability. We have added the results in Appendix B.
>
> (1) Robustness to Natural Corruptions (ImageNet-C)
> Across diverse corruption types and severity levels in ImageNet-C, models trained with soft-label regularization (label smoothing, Mixup, CutMix) generally show lower corruption error than the baseline. This behavior aligns with our representation-space analysis. As shown in our representation-space analysis, soft-label regularization produces features that form tighter and more compact clusters with higher alignment to their class centers. We believe this clustering effect leads to larger margins from decision boundaries on average, which in turn makes features less susceptible to boundary crossings under corruption-induced perturbations.
>
> (2) Black-Box Robustness via Transfer-Based Attacks
> We also evaluate transfer attacks, where adversarial examples crafted on a surrogate model are applied to a target model. Soft-label regularized models consistently exhibit lower transfer attack success rates than the baseline.
>
> This trend mirrors the white-box results and aligns with prior findings that adversarial examples transfer when models share similar vulnerable gradient directions or overlapping adversarial subspaces [1–3]. Since such shared perturbation directions also underlie black-box transferability, we interpret our results in a unified way: soft-label regularization produces features with lower RMS and higher cosine similarity to their class centers, which causes perturbations to move representations more toward the origin than toward decision boundaries. As a result, even when the perturbation is generated on a different model, the adversarial direction makes it less likely for transferred adversarial examples to push features across decision boundaries, leading to reduced black-box attack success rates.
>
> [1] Tramèr et al., “The Space of Transferable Adversarial Examples”, arXiv 2017
> [2] Adam et al., “Reducing Adversarial Example Transferability Using Gradient Regularization”, arXiv 2019
> [3] Yang et al., “TRS: Transferability Reduced Ensemble via Encouraging Gradient Diversity and Model Smoothness”, NeurIPS 2021
>
> **Response to [W2]: The impact of bias magnitude on feature geometry is not sufficiently analyzed.**
>
> Thank you for raising the question of how bias magnitudes influence decision-boundary structure. Although bias terms are difficult to control directly during training, we conducted targeted experiments to assess their impact. By deliberately initializing the classifier biases (and weights) to unusually large values, we observed that the decision regions still converged to cone-shaped structures around the origin as training progressed (Appendix F). Additionally, in standard trained models, the learned bias values in the original high-dimensional representation space remain extremely small (Appendix D.2), indicating that they have negligible effect on boundary geometry. These findings support our claim that the minimum-loss point consistently serves as a stable class center even in the presence of bias terms. While precisely determining the theoretical threshold at which large biases would disrupt cone-shaped regions is challenging, our empirical results show that the bias magnitudes obtained under standard training settings are sufficiently small and therefore do not affect the formation of cone-shaped decision boundaries.

---

> ### Author Response · Authors · 2025-11-20
> **Response to Reviewer uj9a**
>
> **Response to [W4]: The ablation logic is not clearly organized, making it difficult to follow the variable–outcome relationships.**
>
> Thank you for your comment regarding a dedicated ablation section. In our setting, soft-label regularization produces two effects that inherently arise together during training: (i) reduction in feature magnitude and (ii) increase in alignment. Because these effects are tightly coupled outcomes of the same optimization dynamics and are difficult to manipulate independently, constructing a traditional module-wise ablation is very challenging in this setting.
>
> Instead, we structured our empirical analysis around the causal roles of these two effects. Specifically, we show that reduced feature RMS directly relates to improved calibration (Section 4.3), while increased cosine similarity is responsible for enhanced adversarial robustness (Section 4.4). Although these two factors cannot be cleanly separated during training, this mechanism-driven organization serves the role of an ablation by clarifying which representation-space change contributes to each behavioral improvement.
>
> We will revise the paper to more explicitly highlight this structure so that readers do not need to infer it from scattered results.
>
> **Response to [Q1]: Are manual feature scaling and soft-label regularization equivalent in how they reduce feature magnitude, and do they differ in their effects on robustness?**
>
> We appreciate the reviewer’s question. Manual feature scaling and soft-label regularization are not fully equivalent. Our post-hoc scaling experiment was designed for two specific purposes: (i) to empirically validate that cone-shaped decision regions preserve class predictions under radial scaling, and (ii) to isolate the effect of reduced feature magnitude by showing that decreasing RMS alone is sufficient to adjust calibration.
>
> Soft-label regularization, however, induces an additional change that manual scaling does not: it increases the cosine similarity between features and their class centers. As shown in Sections 4.2 and 4.4, this improved alignment is the key factor contributing to enhanced adversarial robustness, whereas manual scaling leaves alignment unchanged and therefore does not yield the same robustness improvements. Thus, feature scaling is useful for isolating the role of magnitude in calibration, but it does not replicate the full set of representation-space changes produced by soft-label regularization.
>
> Response to [Q2]: Does the impact of soft-label regularization on feature distributions depend on the class-to-feature dimensionality ratio?
> Thank you for the insightful question. Yes, the effect of soft-label regularization can depend on the relationship between the number of classes and the feature dimensionality. As discussed in Appendix D.1, when the number of classes becomes much larger than the feature dimensionality, the classifier may no longer form clean cone-shaped decision regions, and deviations from the radial structure can emerge. Moreover, because soft-label regularization makes the feature space more compact by tightening clusters and reducing feature magnitudes, it may effectively change how features are distributed within the available dimensionality. This means that the practical class-to-dimension ratio experienced by the model can shift under regularization, which may influence how strongly these geometric effects appear.
>
> However, identifying the exact ratio at which cone-shaped regions cease to hold is challenging, since direct geometric inspection becomes infeasible in high-dimensional spaces. In the general case where feature dimensionality is comparable or larger than the number of classes, we empirically observe cone-shaped structures in the original representation space (Appendix C and D.2). Therefore, the effects of soft-label regularization described in our study remain reliable under these practical settings.

---

### Official Review · Reviewer_Mpdz · 2025-10-31

**Soundness:** 3
**Presentation:** 4
**Contribution:** 3
**Rating:** 6
**Confidence:** 4

**Summary:**

The paper analyzes why soft-label regularizers—Label Smoothing, Mixup, CutMix—improve calibration and gradient-based adversarial robustness by examining geometry in the representation space (penultimate-layer features). It argues that (i) decision regions are cone-like around the origin; (ii) confidence contours and loss gradients radiate from minimal-loss points; and (iii) soft-label regularization reduces feature norms (RMS) while increasing alignment to class “centers,” jointly explaining better calibration and robustness. Evidence spans CIFAR-10/100 and ImageNet with models including ResNet-50, Swin-T, MobileNetV2, ConvNeXt-T, and some pretrained ViT-B/16 variants (Tabs. 1–6; Figs. 1–7, 13–22).

**Strengths:**

- Clear geometric story (cone-shaped decision regions; outward confidence contours; gradient directions) supported by intuitive 2D/3D visualizations and high-dimensional checks (Fig. 1–2; Tabs. 3–4).
- Consistent empirical patterns across backbones: with soft labels, RMS ↓, cos-sim to class center ↑, ECE improves, and attack success ↓ on ImageNet (Table 2; Figs. 3, 13–21).
- Theoretical lens (Theorem 1) linking soft labels to finite-norm features, connecting feature-scaling with temperature scaling for calibration; simple post-hoc feature scaling sanity check (Fig. 22).

**Weaknesses:**

- Slightly dated framing / ConvNet-centric emphasis. While the experiments do include Swin-T and ViT-B/16, the motivation, related-work framing, and chosen regularizers are largely those historically developed for ConvNets; modern Transformer-first training regimes (large-scale weak supervision, diffusion-style objectives, masked modeling, data mixing specific to ViTs) and Transformer-tailored regularizers are not deeply discussed. This makes the narrative feel a bit ConvNet-era despite the ViT results.
- Limited downstream perspective. The study focuses on classification (calibration/robustness) and does not evaluate fine-tuning to downstream tasks such as detection/segmentation/transfer, where representation-space changes could impact task loss landscapes and robustness differently. (No such finetuning sections or tables are provided.)
- Comparison to Transformer literature could be sharper. The Discussion acknowledges that prior claims of ViT robustness/calibration can confound pretraining/regularization differences, but a systematic head-to-head under contemporary ViT training recipes (e.g., stronger data mixing, augment stacks specific to ViTs) is limited (Table 2 contrasts are helpful but narrow).

**Questions:**

- Transformer-first training: Can you add an experiment suite where ViT models are trained under modern ViT recipes (stronger data mixing/augment stacks used in ViT papers) to verify whether the RMS↓ / cos-sim↑ / ECE↓ patterns—and the robustness story—still hold, and how they compare against ConvNets at matched accuracy and compute? (Extend Table 2 with ViT-B/L under such settings.)
- Downstream fine-tuning: Please report whether the softer, more compact features help or hurt fine-tuned downstream tasks (e.g., COCO detection/segmentation or VTAB-transfer). Even a linear-probe vs fine-tune summary table would clarify practical impact beyond classification.
- Ablations on representation “centers”: Your results favor minimum-loss points over class means/weights (Table 1). Could you quantify how stable these centers are across seeds and training schedules, and whether center estimation noise affects the reported correlations?

---

> ### Author Response · Authors · 2025-11-20
> **Response to Reviewer Mpdz**
>
> We greatly appreciate the positive and encouraging feedback from reviewer Mpdz, noting that our geometric perspective is clear and well supported by visualizations and high-dimensional checks, the empirical patterns across architectures are consistent and meaningful, and the theoretical connection between soft labels, feature norms, and calibration is insightful. We will address all concerns raised by the reviewer and revise the paper accordingly.
>
> **Response to [W1, W3]: The augmentation techniques used in the paper (e.g., Mixup, CutMix) are outdated for Transformer based models.**
>
> We appreciate the suggestion to incorporate augmentation strategies tailored for modern Transformer architectures. Our objective, however, is to ensure that architectural comparisons remain controlled and are not confounded by heterogeneous training recipes. For this reason, we adopt augmentation choices that are contemporary, widely used, and shared across both ViT and CNN pipelines.
>
> Importantly, the use of Mixup (α=1.0) and CutMix (α=1.0) is not a ConvNet-specific convention. Rather, it is a standard data-mixing regime used in recent ViT training practices. The choice α=1.0 corresponds to sampling λ from a uniform distribution, yielding the broadest range of effective mixing ratios. Multiple recent studies in ViT training frameworks [1–5] report that α=1.0 remains the recommended setting, as it promotes stable optimization, mitigates positional embedding variance shifts, and improves generalization performance for Transformer models.
>
> Therefore, our training recipe reflects widely adopted configurations in the ViT literature and ensures that comparisons in Table 2 are made under consistent, architecture-agnostic regularization conditions.
>
> Finally, we would like to emphasize that the goal of this work is to analyze the effect of soft-label based regularizers (e.g., label smoothing, Mixup, and CutMix) on calibration and robustness from a representation-space perspective. Since all three methods share the common mechanism of modifying label distributions, they form a coherent family of regularizers that directly aligns with our research question.
>
> [1] Kim et al., “Configuring Data Augmentations to Reduce Variance Shift in Positional Embedding of Vision Transformers”, AAAI 2025
> [2] Yun et al., “CutMix: Regularization Strategy to Train Strong Classifiers with Localizable Features”, ICCV 2019
> [3] Kosson et al., “Rotational Equilibrium: How Weight Decay Balances Learning Across Neural Networks”, ICML 2024
> [4] Fini et al., “Semi-supervised learning made simple with self-supervised clustering”, CVPR 2023
> [5] Wang et al., “Adapting LLaMA Decoder to Vision Transformer”, arXiv 2024

---

> > ### Comment · Reviewer_Mpdz · 2025-11-25
> > **Comment to the author**
> >
> > Thank you for the clarification! One more question is
> > - Since the proposed method is not restricted to ConvNets, it would be very helpful to include ViT-based results that directly report accuracy.
> > - At the moment, the only ViT-related result I found is in Table 6, which does not directly reflect accuracy, so additional ViT experiments (or clearer accuracy-focused reporting for ViTs) would strengthen the paper.

---

> > > ### Author Response · Authors · 2025-11-27
> > > **Additional Response to Reviewer Mpdz**
> > >
> > > Thank you very much for your thoughtful follow-up questions and for actively engaging in the discussion. Following the reviewer’s request, we have added additional experiments on ViT-B-16 to the revised manuscript. Due to computational resource limitations, we were unable to exhaustively evaluate all regularization settings for the ViT model. Instead, we included two representative configurations: (i) the baseline model without soft-label regularization, and (ii) the CutMix (α = 1.0) setting, which is the most widely used and standard configuration in large-scale ViT training pipelines.
> > >
> > > The newly added ViT-B-16 results are included in:
> > > - Table. 1 (correlation analysis with class center candidates)
> > > - Table. 2 (accuracy, RMS, alignment, calibration, and adversarial robustness)
> > > - Figure. 16 (feature RMS, center-alignment, calibration plots)
> > > - Figure. 22 (robustness under strong PGD and AutoAttack settings)
> > > These newly added results directly report the validation accuracy of ViT-B-16, alongside all other key metrics.
> > >
> > > **Consistency of ViT behavior with our main findings**
> > > ViT-B-16 shows consistent trends aligned with our observations across other architectures (ResNet, Swin, ConvNeXt, and MobileNet):
> > > 1. Reduced feature RMS under soft-label regularization (Table. 2 and top row in Figure. 16).
> > > 2. Improved alignment with class centers via higher cosine similarity (Table. 2 and middle row in Figure. 16).
> > > 3. Mitigated overconfidence and better calibration (Table. 2 and bottom row in Figure. 16).
> > > 4. Enhanced adversarial robustness, with lower FGSM, PGD, and AutoAttack success rates (Table. 2, middle row in Figure. 16, and Figure. 22).
> > >
> > > **Alignment–robustness relationship**
> > > In Section 4.4, better alignment with class centers leads to higher robustness, since perturbations tend to move features toward the origin rather than across decision boundaries. ViT-B-16 follows the same pattern: data with higher alignment corresponds to lower vulnerability to gradient-based attacks (Figures. 16 and 22).
> > >
> > > Thus, the newly added ViT experiments confirm that our representation-space explanation generalizes beyond ConvNets and applies equally well to transformer-based models (ViT, Swin, ConvNeXt).
> > >
> > > Lastly, we clarify that the accuracy, calibration, and robustness results for PyTorch-provided pretrained models that appear in Table. 6 can be found in Table. 8.

---

> ### Author Response · Authors · 2025-11-20
> **Response to Reviewer Mpdz**
>
> **Response to [W2, Q2]: The study on the impact of regularization on downstream task performance is limited.**
>
> Thank you for the suggestion to evaluate the effect of regularization on downstream tasks. In response, we conducted additional transfer-learning experiments using linear probes on CIFAR-100. The results are summarized in the table below. We find that soft-label regularization does not improve downstream performance; in most cases, it slightly degrades linear-probe accuracy compared to the baseline, despite yielding small gains on the original ImageNet task.
>
> | Method           | ResNet-50 | MobileNetV2 | Swin-T | ConvNeXt-T |
> |------------------|---------------------------|------------------------------|-------------------------|-------------------------------|
> | Baseline         | 57.2                      | 51.0                         | 74.1                    | 72.8                          |
> | Label Smoothing  | 53.4                      | 51.4                         | 73.1                    | 70.1                          |
> | Mixup            | 55.0                      | 50.3                         | 74.9                    | 75.0                          |
> | CutMix           | 43.3                      | 42.7                         | 74.0                    | -                             |
>
>
> This observation aligns with recent findings showing that label smoothing increases ImageNet accuracy but reduces linear-probe transfer performance on CIFAR-10 [1]. Their analysis attributes this degradation to reduced transferability of features, as soft-label regularizers suppress informative, discriminative variations that are useful for downstream tasks.
>
> Our representation-space analysis provides a coherent explanation for this phenomenon. As shown in Sec. 4 and Table 2, soft-label regularization reduces the RMS of features and increases their alignment with class centers, producing tighter clusters. While such compact representations help calibration and adversarial robustness, they also reduce the diversity and expressiveness of the learned features. Downstream transfer tasks rely on generalizable, broad representations rather than class-specific collapses. Therefore, we believe the observed degradation in linear-probe accuracy is a natural consequence of the representation-space dynamics induced by soft-label regularization.
>
> [1] Zhou et al., “MaxSup: Overcoming Representation Collapse in Label Smoothing”, NeurIPS 2025
>
> **Response to [Q1]: Comparison between CNN and ViT under matched accuracy and compute.**
>
> Thank you for the insightful suggestion. We agree that a fair comparison between CNN and Transformer architectures requires matching model capacity and compute. For this reason, we selected Swin-T rather than ViT-B, as Swin-T (28.2M params) and ResNet50 (25.5M) have comparable parameter counts and achieve nearly identical ImageNet accuracy (75.8% vs. 76.1%). This provides a controlled setting for isolating the effects of regularization on representation-space geometry.
>
> Under this matched setup, we first observe that both models exhibit the same qualitative response to soft-label regularization. As shown in Table 2: (i) RMS of representation vectors decreases and (ii) cosine similarity to class centers increases for both Swin-T and ResNet50 when applying soft-label regularizers. This confirms that the representation-space phenomena we study, namely movement toward the origin and tighter alignment, are architecture-agnostic and affect both CNNs and ViTs in a consistent manner.
>
> After this shared representation-space change, however, the downstream calibration and robustness outcomes diverge. Despite benefiting from reduced RMS and increased alignment, Swin-T remains more overconfident and more vulnerable to adversarial attacks than ResNet50, even under identical training and compute budgets. This holds across multiple white-box attacks (e.g., FGSM, PGD, AutoAttack) as shown in Table 2, and also under transferred black-box attacks (Table 4).
>
> Thus, although soft-label regularization produces similar representation-space changes in both architectures, Swin-T still shows weaker robustness and calibration than a comparably sized CNN. This indicates that the remaining differences stem from architectural characteristics rather than training settings.

---

> ### Author Response · Authors · 2025-11-20
> **Response to Reviewer Mpdz**
>
> **Response to [Q3]: How stable are the minimum-loss centers across seeds and training setups?**
>
> We appreciate the question regarding the stability of the minimum-loss centers.
>
> 1. Determinism and uniqueness of the centers.
> For a fixed trained classifier, each class center is defined as the minimizer of the cross-entropy loss in the representation space. Since cross-entropy is convex with respect to logits and the logits are affine functions of the representation (Sec. 3.2), the loss is convex in the representation as well. Thus, each class has a unique global minimum. In practice, we locate this point via gradient descent from a fixed initialization (the origin), making the procedure fully deterministic and independent of random seeds.
>
> 2. Ablation on center-estimation hyperparameters.
> To assess sensitivity to estimation noise, we conduct additional experiments on ResNet-50 (ImageNet, baseline) by broadly varying the optimization settings used to locate the minimum-loss centers (Appendix I). Specifically, we used SGD with and without a cosine-annealing learning-rate schedule, tested learning rates of 0.001, 0.01, and 0.1, and varied the number of gradient-descent update steps across 50, 100, and 200 iterations. Despite this wide sweep of hyperparameters (18 configurations in total), the resulting Pearson correlations between confidence and cosine similarity to the estimated centers remained highly consistent, ranging only from 0.5353 to 0.5452. This narrow variation indicates that the estimation process is very stable and that center-estimation noise does not materially influence the reported correlations.

---

### Author Response · Authors · 2025-11-20
**General Response by Authors**

We sincerely thank all reviewers for their thoughtful and constructive feedback. Several reviewers highlighted the strengths of our work, including the clarity of the geometric perspective connecting soft-label regularization to representation-space behavior (Mpdz, uj9a, CRdf), the combination of intuitive 2D/3D visualizations with quantitative high-dimensional analyses (Mpdz, guCA), the comprehensive experimental evaluation (Mpdz, guCA, uj9a), and the novelty of resolving the contradiction between reduced confidence and improved robustness (guCA). We are grateful for these encouraging assessments.
In response to reviewer suggestions, we have expanded the manuscript to include several new analyses and clarifications:

- Discussion on the generalizability of geometric insights and future work (Section 8)
- ViT-B-16 results (Tables 1, 2 and Figures 17, 23)
- Robustness to natural corruptions (ImageNet-C) (Appendix B.1)
- Black-box adversarial transferability analysis (Appendix B)
- Hyperparameter sensitivity of minimum-loss center estimation (Appendix I)
- Strengthening the proof of Theorem 1 to make it fully agnostic to the presence of bias terms (Appendix J)
- Effect of soft-label regularization on downstream transfer performance (Appendix M)
- PGD trajectory analysis demonstrating how soft-label regularization influences feature movement under iterative attacks (Appendix N)

We have uploaded the first revision of the manuscript (changes are highlighted by the red color).
We hope our response and revision sincerely address all the reviewers’ concerns.

---

### Meta-Review · Area_Chair_xUzT · 2026-01-08

**Summary:**

The paper investigates how existing soft-label based regularization techniques, e.g., label smoothing, mixup, etc., improve model calibration and adversarial robustness. Reviewer scores are quite divergent; two reviewers recommended marginal acceptance (6, 6), while the other two leaned toward rejection (2, 4). After reading the rebuttal, AC finds that several concerns remain outstanding and valid, including limited practical significance, limited generality of the claims to other domains, and insufficient theoretical support, etc. AC agrees with these concerns and recommends rejection for the current submission.

**Reviewer Concerns:**

The major concerns raised from the negative reviewers (CRdf - 2, guCA - 4) can be summarized as the following:

- Lack of mathematical rigor (CRdf)
- Weak causal support for robustness claims (CRdf)
- Effectiveness under strong attacks (CRdf)
- Limited practical significance (CRdf)
- Theoretical analysis relying on simplifying assumptions (guCA)
- Generality of the geometric assumptions to other domains (guCA)

AC agrees with the concerns regarding the theoretical analysis, particularly that the paper would benefit from more rigorous support under milder assumptions. The concern regarding practical significance raised by CRdf remains outstanding and is considered crucial. The generality concern raised by guCA also remains unresolved; AC believes that addressing this issue is important for broadening the paper’s audience. In addition, AC agrees with Mpdz that the claims should be tested more extensively on Transformers, even after the rebuttal.

**Reviewer Scores:**

- Reviewer Mpdz: Initially 6. Would keep his/her original score.
- Reviewer uj9a: Initially 6. Would keep his/her original score.
- Reviewer CRdf: Initially 2. Would keep his/her original score.
- Reviewer guCA: Initially 4. Would keep his/her original score.

---

### Decision · Program_Chairs · 2026-01-26

Reject